# Testing Semantic Importance via Betting

**Jacopo Teneggi**
Johns Hopkins University
`jtenegg1@jhu.edu`

**Jeremias Sulam**
Johns Hopkins University
`jsulam1@jhu.edu`

## Abstract

Recent works have extended notions of feature importance to *semantic concepts* that are inherently interpretable to the users interacting with a black-box predictive model. Yet, precise statistical guarantees such as false positive rate and false discovery rate control are needed to communicate findings transparently, and to avoid unintended consequences in real-world scenarios. In this paper, we formalize the global (i.e., over a population) and local (i.e., for a sample) statistical importance of semantic concepts for the predictions of opaque models by means of conditional independence, which allows for rigorous testing. We use recent ideas of sequential kernelized independence testing to induce a rank of importance across concepts, and we showcase the effectiveness and flexibility of our framework on synthetic datasets as well as on image classification using several vision-language models.

## 1 Introduction

Providing guarantees on the decision-making processes of autonomous systems, often based on complex black-box machine learning models, is paramount for their safe deployment. This need motivates efforts towards *responsible artificial intelligence*, which broadly entails questions of reliability, robustness, fairness, and interpretability. One popular approach to the latter is to use post-hoc explanation methods to identify the features that contribute the most towards the predictions of a model. Several alternatives have been proposed over the past few years, drawing from various definitions of *features* (e.g., pixels—or groups thereof—for vision tasks [41], words for language tasks [21], or nodes and edges for graphs [84]) and of *importance* (e.g, gradients for Grad-CAM [57], Shapley values for SHAP [7, 15, 42, 70], or information-theoretic quantities [43]). While most explanation methods highlight important features in the input space of the predictor, users may care more about their meaning. For example, a radiologist may want to know whether a model considered the size and spiculation of a lung nodule to quantify its malignancy, and not just its raw pixel values.

To decouple importance from input features, Kim et al. [34] showed how to learn the vector representation of *semantic concepts* that are inherently interpretable to users (e.g., "stripes", "sky", or "sand") and how to study their gradient importance for model predictions. Recent vision-language (VL) models that jointly learn an image and text encoder, such as CLIP [16, 52], have made these representations—commonly referred to as concept activation vectors (CAVs)—more easily accessible. With these models, obtaining the representation of a concept boils down to a forward pass of the pretrained text encoder, which alleviates the need of a dataset comprising images annotated with their concepts. Several recent works have defined semantic importance—both with CAVs and VL models—by means of *concept bottleneck models* (e.g., CBM [36], PCBM [85], LaBo [83]), information-theoretic quantities (e.g., V-IP [37]), sparse coding (e.g., CLIP-IP-OMP [13], SpLiCe [6]), *network dissection* [2] (e.g., CLIP-DISSECT [47], TextSpan [26], INViTE [14]), or causal inference (e.g., DiConStruct [44], Sani et al. [55]).

On the other hand, it is important to communicate findings of important features precisely and transparently in order to avoid unintended consequences in downstream decision tasks. Going back to the example of a radiologist diagnosing lung cancer, how should they interpret two concepts with

38th Conference on Neural Information Processing Systems (NeurIPS 2024).

different importance scores? Does their difference in importance carry any statistical meaning? To start addressing similar questions [10, 71] introduced statistical tests for the local (i.e., on a sample) conditional independence structure of a model's predictions. Framing importance by means of conditional independence allows for rigorous testing with false positive rate control. That is, for a user-defined significance level $\alpha \in (0, 1)$, the probability of wrongly deeming a feature important is no greater than $\alpha$, which directly conveys the uncertainty in an explanation. Yet these methods consider features as coordinates in the input space, and it is unclear how to extend these ideas to abstract, semantic concepts.

In this work, we formalize semantic importance at three distinct levels of statistical independence with null hypotheses of increasing granularity: $(i)$ marginally over a population (i.e., global importance), $(ii)$ conditionally over a population (i.e., global conditional importance), and $(iii)$ for a sample (i.e., local conditional importance).[1] Each of these notions will allow us to inquire the extent to which the output of a model depends on specific concepts—both over a population and on specific samples—and thus deem them important. To test for these notions of semantic importance, instead of classical (or *offline* [58]) independence testing techniques [5, 10, 11, 28, 29, 69, 86], which are based on $p$-values and informally follow the rule *"reject if $p \leq \alpha$"*, we propose to use principles of *testing by betting* (or *sequential* testing) [59], which are based on $e$-values [76] and follow the *"reject when $e \geq 1/\alpha$"* rule. As we will expand on, this choice is motivated by the fact that sequential tests are data-efficient and adaptive to the hardness of the problem—which naturally induces a rank of importance. We will couple principles of conditional randomization testing (CRT) [11] with recent advances in sequential kernelized independence testing (SKIT) [51, 62], and introduce two novel procedures to test for our definitions of semantic importance: the conditional randomization SKIT (C-SKIT) to study global conditional importance, and—following the explanation randomization test (XRT) framework [71]—the explanation randomization SKIT (X-SKIT) to study local conditional importance. We will illustrate the validity of our proposed tests on synthetic datasets, and showcase their flexibility on zero-shot image classification on real-world datasets across several and diverse VL models.

## 1.1 Summary of Contributions and Related Works

In this paper, we will rigorously define notions of statistical importance of semantic concepts for the predictions of black-box models via conditional independence—both globally over a population and locally for individual samples. For any set of concepts, and for each level of independence, we introduce novel sequential testing procedures that induce a rank of importance. Before presenting the details of our methodology, we briefly expand on a few distinctive features of our work.

**Explaining nonlinear predictors.** Compared to recent methods based on concept bottleneck models [36, 83, 84], our framework does not require training a surrogate linear classifier because we study the semantic importance structure of any given, potentially nonlinear and randomized model. This distinction is not minor—training concept bottleneck models results on explanations that pertain to the surrogate (linear) model instead of the original (complex, nonlinear) predictor, and these simpler surrogate models typically reduce performance [48, 85]. In contrast, we provide statistical guarantees directly on the original predictor that would be deployed in the wild.

**Flexible choice of concepts.** Furthermore, our framework does not rely on the presence of a large concept bank (but it can use one if it is available). Instead, we allow users to directly specify which concepts they want to test. This feature is important in settings that involve diverse stakeholders. In medicine, for example, there are physicians, patients, model developers, and members of the regulatory agency tasked with auditing the model—each of whom might prefer different semantics for their explanations. Current explanation methods cannot account for these differences off-the-self.

**Local semantic explanations.** Our framework entails explanations for specific (fixed) inputs, whereas prior approaches that rely on the weights of a linear model only inform on global notions of importance. Recently, Shukla et al. [64] and Pham et al. [50] set forth ideas of local semantic importance by combining LIME [54] with T-CAV [34], and by leveraging prototypical part networks [46], respectively. Our work differs in that it does not apply to images only, it considers formal notions of statistical importance rather than heuristics of gradient importance, and it provides guarantees such as Type I error and false discovery rate (FDR) control.

---

[1]We adopt the distinction between *global* and *local* importance as presented in [18].

**Sequential kernelized testing.** Motivated by the statistical properties of kernelized independence tests [62], we will employ the maximum mean discrepancy (MMD) [29] as the test statistic in our proposed procedures. The recent related work in Shaer et al. [58] introduces the sequential version of the conditional randomization test (CRT) [11], dubbed e-CRT because of the use of $e$-values. Unlike our work, Shaer et al. [58] employ residuals of a predictor as test statistic, they do so in the context of global tests only, and unrelated to questions of semantic interpretability.

## 2 Background

In this section, we briefly introduce the necessary notation and general background. Throughout this work, we will denote random variables with capital letters, and their realizations with lowercase. For example, $X \sim P_X$ is a random variable sampled from $P_X$, and $x$ indicates an observation of $X$.

**Problem setup.** We consider $k$-fold classification problems such that $(X, Y) \sim P_{XY}$ is a random sample $X \in \mathcal{X}$ with its one-hot label $Y \in \{0, 1\}^k$, and $(x, y)$ denotes a particular observation. We assume we are given a fixed predictive model, consisting of an encoder $f : \mathcal{X} \to \mathbb{R}^d$ and a classifier $g : \mathbb{R}^d \to \mathbb{R}^k$ such that $h = f(x)$ is a $d$-dimensional representation of $x$, and $\hat{y}_{k'} = g(h)_{k'} = g(f(x))_{k'}$ is the prediction of the model for a particular class $k'$ (e.g., $\hat{y}_{k'}$ is the output, or score, for class "dog"). Naturally, $H, \hat{Y}$ denote the random counterparts of $h$ and $\hat{y}$. Although our contributions do not make any assumptions on the performance of the model, $f$ and $g$ can be thought of as *good* predictors, e.g. those that approximate the conditional expectation of $Y$ given $X$.

**Concept bottleneck models (CBMs).** Let $c = [c_1, \ldots, c_m] \in \mathbb{R}^{d \times m}$ be a dictionary of $m$ concepts such that $\forall j \in [m] := \{1, \ldots, m\}$, $c_j \in \mathbb{R}^d$ is the representation of the $j^{\text{th}}$ concept—either obtained via CAVs [34] or a VL model's text encoder. Then, $z = \langle c, h \rangle$ is the projection of the embedding $h$ onto the concepts $c$, and, with appropriate normalization, $z_j \in [-1, 1]$ is the *amount* of concept $c_j$ in $h$. Intuitively, CBMs project dense representations onto the subspace of interpretable semantic concepts [36], and their performance strongly depends on the size of the dictionary [48]. For example, it is common for $m$ to be as large as the embedding size (e.g., $d = 768$ for CLIP:ViT-L/14). In this work, instead, we let concepts be user-defined, allowing for cases where $m \ll d$ (e.g., $m = 20$). This is by design as $(i)$ the contributions of this paper apply to any set of concepts, and $(ii)$ it has been shown that humans can only gain valuable information if semantic explanations are succinct [53]. However, we remark that the construction of informative concept banks—especially for domain-specific applications—is subject of ongoing complementary research [20, 48, 78, 80, 81].

**Conditional randomization tests.** Recall that two random variables $A, B$ are conditionally independent if, and only if, given a third random variable $C$, it holds that $P_{A|B,C} = P_{A|C}$ (i.e., $A \perp\!\!\!\perp B \mid C$). That is, $B$ does not provide any more information about $A$ with $C$ present. Candes et al. [11] introduced the conditional randomization test (CRT), based on the observation that if $A \perp\!\!\!\perp B \mid C$, then the triplets $(A, B, C)$ and $(A, \widetilde{B}, C)$ with $\widetilde{B} \sim P_{B|C}$, are exchangeable. That is, $P_{ABC} = P_{A\widetilde{B}C}$ and one can essentially *mask* $B$ without changing the joint distribution. Opposite to classical methods, the CRT assumes the conditional distribution of the covariates is known (i.e., $P_{B|C}$), which lends itself to settings with ample unlabeled data.

With this general background, we now present the main technical contributions of this paper.

## 3 Testing Semantic Importance via Betting

Our objective is to test the statistical importance of semantic concepts for the predictions of a fixed, potentially nonlinear model, while inducing a rank of importance. Fig. 1 depicts the problem setup—a fixed model, composed of the encoder $f$ and classifier $g$, is probed via a set of concepts $c$. This figure also illustrates the key difference with post-hoc concept bottleneck models (PCBMs) [85], in that we do not train a sparse linear layer to approximate $\mathbb{E}[Y \mid Z]$. Instead, we focus on characterizing the dependence structure between $\hat{Y}$ and $Z$. Herein, we will drop the $\hat{y}_{k'}$ notation and simply write $\hat{y}, \hat{Y}$ because we always consider the output of the model for a particular class individually.

### 3.1 Formalizing Statistical Importance of Semantic Concepts

We start by defining *global semantic importance* as marginal dependence between $\hat{Y}$ and $Z_j$, $j \in [m]$.

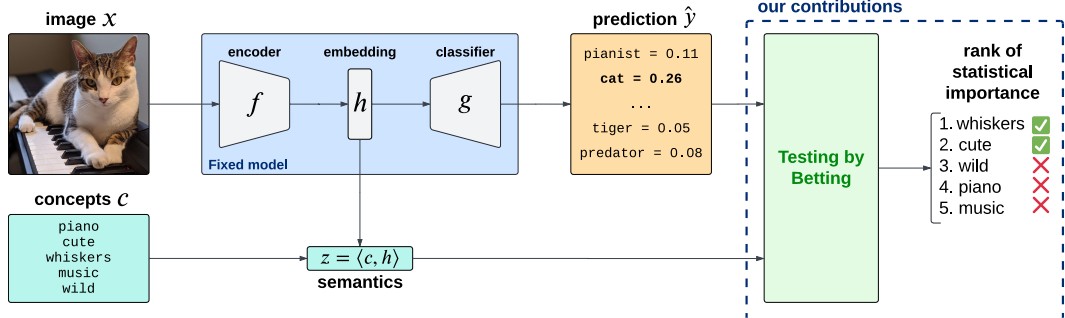

Figure 1: Overview of the problem setup and our contribution.

**Definition 1** (Global semantic importance). For a concept $j \in [m]$,

$$H_{0,j}^{\mathrm{G}} : \ \hat{Y} \perp\!\!\!\perp Z_j \tag{1}$$

is the global semantic independence null hypothesis.

Rejecting $H_{0,j}^{\mathrm{G}}$ means that we have observed enough evidence to believe the response of the model depends on concept $j$, i.e. concept $j$ is globally important over the population. Note that both $\hat{Y}$ and $Z_j$ are fixed functions of the same random variable $H$, i.e. $\hat{Y} = g(H)$ and $Z_j = \langle c_j, H \rangle$. Then, it is reasonable to wonder whether there is any point in testing $H_{0,j}^{\mathrm{G}}$ at all—can we obtain two independent random variables from the same one? For example, let $g$ be a linear classifier such that $\hat{Y} = \langle w, H \rangle$, $w \in \mathbb{R}^d$. Intuition might suggest that if $\langle w, c_j \rangle = 0$ then $\hat{Y} \perp\!\!\!\perp Z_j$, i.e. if the classifier is orthogonal to a concept, then the concept cannot be important. We show in the short lemma below (whose proof is in Appendix C.1) that this is false, and that, arguably unsurprisingly, statistical independence is different from orthogonality between vectors, which motivates the need for our testing procedures.

**Lemma 1.** *Let $\hat{Y} = \langle w, H \rangle$, $w \in \mathbb{R}^d$. If $d \geq 3$, then $H_{0,j}^{G}$ is true $\overset{\Longleftarrow}{\not\Longrightarrow} \langle w, c_j \rangle = 0$.*

The null hypothesis $H_{0,j}^{\mathrm{G}}$ above precisely defines the global importance of a concept, but it ignores the information contained in the rest of them, and concepts may be correlated. For example, predictions for class "dog" may be independent of "stick" given "tail" and "paws", although "stick" is marginally important. To address this, and inspired by [11], we define *global conditional semantic importance*.

**Definition 2** (Global conditional semantic importance). For a concept $j \in [m]$, let $-j := [m] \setminus \{j\}$ denote all but the $j^{\mathrm{th}}$ concept. Then,

$$H_{0,j}^{\mathrm{GC}} : \ \hat{Y} \perp\!\!\!\perp Z_j \mid Z_{-j} \tag{2}$$

is the global conditional semantic independence null hypothesis.

Analogous to Definition 1, rejecting $H_{0,j}^{\mathrm{GC}}$ means that we have accumulated enough evidence to believe the response of the model depends on concept $j$ even in the presence of the remaining concepts, i.e. there is information about $\hat{Y}$ in concept $j$ that is missing from the rest.

We stress an important distinction between Definition 2 and PCBMs: the latter approximate $\mathbb{E}[Y \mid Z]$ with a sparse linear layer, which is *inherently interpretable* because the regression coefficients directly inform on the global conditional independence structure of the predictions, i.e. if $\hat{Y} = \langle \beta, Z \rangle$, $\beta \in \mathbb{R}^m$ then $\hat{Y} \perp\!\!\!\perp Z_j \mid Z_{-j} \iff \beta_j = 0$. In this work, however, we do not assume any parametrization between the concepts and the labels because we want to provide guarantees on the original, fixed classifier $g$ that acts directly on an embedding $h$. From this conditional independence perspective, PCBMs can be interpreted as a (parametric) test of true linear independence (i.e., $H_{0,j}^{\mathrm{PCBM}} : Y \perp\!\!\!\perp Z_j \mid Z_{-j}$) between the concepts and the labels (note that $H_{0,j}^{\mathrm{PCBM}}$ has the *true* label $Y$ and not the prediction $\hat{Y}$), whereas we study the semantic structure of the predictions of a complex model, which may have learned spurious, non-linear correlations of these concepts from data.

Akin to the CRT [11], we assume we can sample from the conditional distribution of the concepts, i.e. $P_{Z_j \mid Z_{-j}}$. Within the scope of this work, $m$ is small ($m \approx 20$) and we will show how to effectively approximate this distribution with nonparametric methods that do not require prohibitive computational resources. This is an advantage of testing *a few* semantic concepts compared to input

features, especially for imaging tasks where the number of pixels is large ($\gtrsim 10^5$) and learning a conditional generative model (e.g., a diffusion model [67]) may be expensive.

Finally, we define the notion of *local conditional semantic importance*. That is, we are interested in finding the most important concepts for the prediction of the model locally on a particular input $x$, i.e. $\hat{y} = g(f(x))$. Recently, [10, 71] showed how to deploy ideas of conditional randomization testing for local explanations of machine learning models. Briefly, let $B, C$ be random variables and $\eta(B, C)$ a fixed, possibly randomized, real-valued predictor. For an observation $(b, c)$, the explanation randomization test (XRT) [71] null hypothesis is $\eta(b, c) \overset{d}{=} \eta(\widetilde{B}, c)$, $\widetilde{B} \sim P_{B|C=c}$. That is, the *observed* value of $B$ does not affect the distribution of the response given the *observed* value of $C$. We now generalize these ideas.

**Definition 3** (Local conditional semantic importance). For a fixed $z \in [-1, 1]^m$ and any $C \subseteq [m]$, denote $\hat{Y}_C = g(\widetilde{H}_C)$ with $\widetilde{H}_C \sim P_{H|Z_C=z_C}$. Then, for a concept $j \in [m]$ and a subset $S \subseteq [m] \setminus \{j\}$,

$$H_{0,j,S}^{\text{LC}} : \ \hat{Y}_{S\cup\{j\}} \overset{d}{=} \hat{Y}_S \tag{3}$$

is the local conditional semantic independence null hypothesis.

In words, rejecting $H_{0,j,S}^{\text{LC}}$ means that, given the observed concepts in $S$, concept $j \notin S$ affects the distribution of the response of the model, hence it is important. For this test, we assume we can sample from the conditional distribution of the embeddings given a subset of concepts (i.e., $P_{H|Z_C=z_C}$). This is equivalent to solving an inverse problem stochastically, since $z = \langle c, h \rangle$ and $c$ is not invertible ($c \in \mathbb{R}^{d \times m}$, $m \ll d$). Hence, there are several embeddings $h$ that could have generated the observed $z_C$. We will use nonparametric sampling ideas to achieve this, stressing that it suffices to sample the embeddings $H$ and not an entire input image $X$ since the classifier $g$ directly acts on $h$ and the encoder $f$ is deterministic. Finally, we remark that $H_{0,j,S}^{\text{LC}}$ differs from the XRT null hypothesis in that conditioning is performed in the space of semantic concepts instead of the input's.

With these precise notions of semantic importance, we now show how to test for each one of them with principles of sequential kernelized independence testing (SKIT) [51].

## 3.2 Testing by Betting

A classical approach to hypothesis testing consists of formulating a null hypothesis $H_0$, collecting data, and then summarizing evidence by means of a $p$-value. Under the null, the probability of observing a small $p$-value is small. Thus, for a significance level $\alpha \in (0, 1)$, we can reject $H_0$ if $p \leq \alpha$. In this setting, all data is collected first, and then processed later (i.e., offline).

Alternatively, one can instantiate a game between a bettor and nature [59, 60]. At each turn of the game, the bettor places a wager against $H_0$, and then nature reveals the truth. If the bettor wins, they will increase their wealth, otherwise lose some. More formally, and as is commonly done [51, 58, 62], we define a wealth process $\{K_t\}_{t \in \mathbb{N}_0}$ with $K_0 = 1$ and $K_t = K_{t-1} \cdot (1 + v_t \kappa_t)$ where $v_t, \kappa_t \in [-1, 1]$ are a betting fraction and the payoff of the bet, respectively. It is now easy to see that when $v_t \kappa_t \geq 0$ (i.e., the bettor wins) the wealth increases, and the opposite otherwise. If the payoff $\kappa_t$ guarantees the game is *fair*, i.e. the bettor cannot accumulate wealth under the null, then we can use the wealth process to reject $H_0$ with Type I error control (details in Appendix A). In particular, for a significance level $\alpha \in (0, 1)$, we denote $\tau := \min\{t \geq 1 : K_t \geq 1/\alpha\}$ the *rejection time* of $H_0$.

The choice of using sequential testing is motivated by two fundamental properties. First, sequential tests are *adaptive* to the hardness of the problem, sometimes provably [62, Proposition 3]. That is, the harder it is to reject the null, the longer the test will take, and vice versa. This naturally induces a rank of importance across concepts—if concept $c_j$ rejects faster than $c_{j'}$, then $c_j$ is more important (i.e., it is easier to reject the null hypothesis that the predictions do not depend on $c_j$). We stress that this is not always possible by means of $p$-values because they do not measure effect sizes: consider two concepts that reject their respective nulls at the same significance level; one cannot distinguish which—if any—is more important. As we will show in our experiments, all tests used in this work are adaptive in practice, but statistical guarantees on their rejection times are currently open questions, and we consider them as future work. Second, sequential tests are *sample-efficient* because they only analyze the data is needed to reject, which is especially important for conditional randomization tests. In the offline scenario, we would have to resample the entire dataset several times (which is expensive), but the sequential test would terminate in at most the size of the dataset [24].

| **Algorithm 1** Level-$\alpha$ C-SKIT for concept $j$ | **Algorithm 2** Level-$\alpha$ X-SKIT for concept $j$ |
|---|---|
| **Input:** Stream $(\hat{Y}^{(t)}, Z_j^{(t)}, Z_{-j}^{(t)}) \sim P_{\hat{Y} Z_j Z_{-j}}$. | **Input:** Observation $z$, subset $S \subseteq [m] \setminus \{j\}$. |
| 1: $K_0 \leftarrow 1$ | 1: $K_0 \leftarrow 1$ |
| 2: Initialize ONS strategy (Algorithm A.1) | 2: Initialize ONS strategy (Algorithm A.1) |
| 3: **for** $t = 1, \ldots$ **do** | 3: **for** $t = 1, \ldots$ **do** |
| 4:     Compute $\rho_t$ as in Eq. (4) | 4:     Compute $\rho_t$ as in Eq. (5) |
| 5:     $D^{(t)} = (\hat{Y}^{(t)}, Z_j^{(t)}, Z_{-j}^{(t)})$ | 5:     Sample $\widetilde{H}_{S \cup \{j\}}^{(t)} \sim P_{H \mid Z_{S \cup \{j\}} = z_{S \cup \{j\}}}$ |
| 6:     Sample $\widetilde{Z}_j^{(t)} \sim P_{Z_j \mid Z_{-j} = Z_{-j}^{(t)}}$ | 6:     Sample $\widetilde{H}_S^{(t)} \sim P_{H \mid Z_S = z_S}$ |
| 7:     $\widetilde{D}^{(t)} \leftarrow (\hat{Y}^{(t)}, \widetilde{Z}_j^{(t)}, Z_{-j}^{(t)})$ | 7:     $\hat{Y}_{S \cup \{j\}}^{(t)} \leftarrow g(\widetilde{H}_{S \cup \{j\}}^{(t)}), \hat{Y}_S^{(t)} \leftarrow g(\widetilde{H}_S^{(t)})$ |
| 8:     $\kappa_t \leftarrow \tanh(\rho_t(D^{(t)}) - \rho_t(\widetilde{D}^{(t)}))$ | 8:     $\kappa_t \leftarrow \tanh(\rho_t(\hat{Y}_{S \cup \{j\}}^{(t)}) - \rho_t(\hat{Y}_S^{(t)}))$ |
| 9:     $K_t \leftarrow K_{t-1} \cdot (1 + v_t \kappa_t)$ | 9:     $K_t \leftarrow K_{t-1} \cdot (1 + v_t \kappa_t)$ |
| 10:     **if** $K_t \geq 1/\alpha$ **then** | 10:     **if** $K_t \geq 1/\alpha$ **then** |
| 11:       **return** $t$ | 11:       **return** $t$ |
| 12:     **end if** | 12:     **end if** |
| 13:     $v_{t+1} \leftarrow$ ONS step | 13:     $v_{t+1} \leftarrow$ ONS step |
| 14: **end for** | 14: **end for** |

### 3.3 Testing Global Semantic Importance with SKIT

Podkopaev et al. [51] show how to design sequential kernelized tests of independence (i.e., $H_0 : A \perp\!\!\!\perp B$) by framing them as particular two-sample tests of the form $H_0 : P = \widetilde{P}$, with $P = P_{AB}$ and $\widetilde{P} = P_A \times P_B$. Similarly to [58, 62], they propose to leverage a simple yet powerful observation about the *symmetry* of the data under $H_0$ [51, Section 4]. We state here the main result we will use in this paper (the proof is included in Appendix A.2).

**Lemma 2** (See [51, 58, 62]). $\forall t \geq 1$, let $X \sim P$ and $\widetilde{X} \sim \widetilde{P}$, and let $\rho_t : \mathcal{X} \rightarrow \mathbb{R}$ be any fixed real-valued function on $\mathcal{X}$. Then, $\kappa_t = \tanh(\rho_t(X) - \rho_t(\widetilde{X}))$ provides a fair game for $H_0 : P = \widetilde{P}$.

That is, Lemma 2 prescribes how to construct valid payoffs for two-samples tests and, consequently, tests of independence. We note that the choice of $\tanh$ provides $\kappa_t \in [-1, 1]$, but any arbitrary anti-symmetric function can be used (e.g., sign). Furthermore, any fixed function $\rho_t$ is valid but, in general, this function should have a positive value under the alternative in order for the bettor to increase their wealth and the testing procedure to have good power.

Going back to the problem studied in this work, note that the global semantic importance null hypothesis $H_{0,j}^{\mathrm{G}}$ in Definition 1 can be directly rewritten as a two-sample test, i.e. $H_{0,j}^{\mathrm{G}} : \hat{Y} \perp\!\!\!\perp Z_j$ is equivalent to $H_{0,j}^{\mathrm{G}} : P_{\hat{Y} Z_j} = P_{\hat{Y}} \times P_{Z_j}$. We follow [51] and use the maximum mean discrepancy (MMD) [29] to measure the distance between the joint and the product of marginals. In particular, let $\mathcal{R}_{\hat{Y}}, \mathcal{R}_{Z_j}$ be two reproducing kernel Hilbert spaces (RKHSs) on the domains of $\hat{Y}$ and $Z_j$, respectively (recall that $\hat{Y}$ and $Z_j$ are univariate). Then, $\rho_t^{\mathrm{SKIT}}$ is the plug-in estimate of the *witness function* of $\mathrm{MMD}(P_{\hat{Y} Z_j}, P_{\hat{Y}} \times P_{Z_j})$ at time $t$.[2] We include the SKIT algorithm and technical details on computing $\rho_t^{\mathrm{SKIT}}$ and $k_t^{\mathrm{SKIT}}$ in Appendix B.1.

**Computational complexity of SKIT.** Analogous to the original presentation in Shekhar and Ramdas [62], the computational complexity of Algorithm B.1 is $\mathcal{O}(\tau^2)$, where $\tau$ is the random rejection time.

We now move on to presenting two novel testing procedures: the conditional randomization SKIT (C-SKIT) for $H_{0,j}^{\mathrm{GC}}$, and the explanation randomization SKIT (X-SKIT) for $H_{0,j,S}^{\mathrm{LC}}$.

### 3.4 Testing Global Conditional Semantic Importance with C-SKIT

Analogous to the discussion in the previous section, we rephrase the global conditional null hypothesis $H_{0,j}^{\mathrm{GC}}$ in Definition 2 as a two sample test $H_{0,j}^{\mathrm{GC}} : P_{\hat{Y} Z_j Z_{-j}} = P_{\hat{Y} \widetilde{Z}_j Z_{-j}}$, $\widetilde{Z}_j \sim P_{Z_j \mid Z_{-j}}$. In contrast with other kernel-based notions of distance between conditional distributions [49, 63, 66]—and akin to the CRT [11]—we assume we can sample from $P_{Z_j \mid Z_{-j}}$, which allows us to directly estimate

---

[2]Recall that $\mathrm{MMD}(P_{AB}, P_A \times P_B)$ is the Hilbert-Schmidt Independence Criterion (HSIC) [28].

$\text{MMD}(P_{\hat{Y}Z_jZ_{-j}}, P_{\hat{Y}\tilde{Z}_jZ_{-j}})$ in our testing procedure (we will expand on how to sample from this distribution shortly). Let $\mathcal{R}_{\hat{Y}}, \mathcal{R}_{Z_j}, \mathcal{R}_{Z_{-j}}$ be three RKHSs on the domains of $\hat{Y}, Z_j$, and $Z_{-j}$ (i.e., $\mathbb{R}, \mathbb{R}, \mathbb{R}^{m-1}$, where $m$ is the number of concepts). Then, at time $t$, the C-SKIT payoff function is

$$\rho_t^{\text{C-SKIT}} := \hat{\mu}_{\hat{Y}Z_jZ_{-j}}^{(t-1)} - \hat{\mu}_{\hat{Y}\tilde{Z}_jZ_{-j}}^{(t-1)}, \tag{4}$$

where $\hat{\mu}_{\hat{Y}Z_jZ_{-j}}^{(t-1)}, \hat{\mu}_{\hat{Y}\tilde{Z}_jZ_{-j}}^{(t-1)}$ are the mean embeddings of their respective distributions in $\mathcal{R}_{\hat{Y}} \otimes \mathcal{R}_{Z_j} \otimes \mathcal{R}_{Z_{-j}}$, and $\otimes$ is the tensor product (see Appendix B.2 for technical details). Algorithm 1 summarizes the C-SKIT procedure, which provides Type I error control for $H_{0,j}^{\text{GC}}$, as we briefly state in the following proposition (see Appendix C.2 for the proof).

**Proposition 1.** $\forall t \geq 1$, let $(\hat{Y}, Z_j, Z_{-j}) \sim P_{\hat{Y}Z_jZ_{-j}}$ and $(\hat{Y}, \tilde{Z}_j, Z_{-j}) \sim P_{\hat{Y}\tilde{Z}_jZ_{-j}}, \tilde{Z}_j \sim P_{Z_j|Z_{-j}}$. Then, $\kappa_t := \tanh(\rho_t^{\text{C-SKIT}}(\hat{Y}, Z_j, Z_{-j}) - \rho_t^{\text{C-SKIT}}(\hat{Y}, \tilde{Z}_j, Z_{-j}))$ provides a fair game for $H_{0,j}^{GC}$.

**Computational complexity of C-SKIT.** First note that $Z_{-j}$ is an $(m-1)$-dimensional vector (where $m$ is the number of concepts). So, at each step of the test, the evaluation of the kernel associated with $\mathcal{R}_{Z_{-j}}$ requires an additional sum over $\mathcal{O}(m)$ terms. Furthermore, C-SKIT needs access to samples from $P_{Z|Z_{-j}}$, and we conclude that the computational complexity of Algorithm 1 is $\mathcal{O}(T_n m \tau^2)$, where $T_n$ represents the cost of the sampler on $n$ samples, and it depends on implementation. For example, in the following, we will use non-parametric samplers with $T_n = \mathcal{O}(n^2)$. Other choices of samplers, such as variational-autoencoders, may have constant cost (e.g., they are trained once and only used for inference).

### 3.5 Testing Local Conditional Semantic Importance with X-SKIT

Attentive readers will have noticed that the local conditional semantic null hypothesis $H_{0,j,S}^{\text{LC}}$ in Definition 3 is already a two-sample test where the test statistic $P$ is the distribution of the response of the model *with* the observed amount of concept $j$ (i.e., $\hat{Y}_{S\cup\{j\}} = g(\tilde{H}_{S\cup\{j\}})$), and the null distribution $\tilde{P}$ *without* (i.e., $\hat{Y}_S = g(\tilde{H}_S)$). Herein, we assume we can sample from $\tilde{H}_C \sim P_{H|Z_C=z_C}$ for any subset $C \subseteq [m]$, i.e. the conditional distribution of dense embeddings with specific concepts, which we will address via nonparametric methods. Then, for an RKHS $\mathcal{R}_{\hat{Y}}$, the X-SKIT payoff function is

$$\rho_t^{\text{X-SKIT}} := \hat{\mu}_{\hat{Y}_{S\cup\{j\}}}^{(t-1)} - \hat{\mu}_{\hat{Y}_S}^{(t-1)} \tag{5}$$

with $\hat{\mu}_{\hat{Y}_{S\cup\{j\}}}^{(t-1)}, \hat{\mu}_{\hat{Y}_S}^{(t-1)}$ mean embeddings of the distributions in $\mathcal{R}_{\hat{Y}}$. That is, $\rho_t^{\text{X-SKIT}}$ is the plug-in estimate of the witness function of $\text{MMD}(\hat{Y}_{S\cup\{j\}}, \hat{Y}_S)$—technical details are in Appendix B.3. Then, the X-SKIT testing procedure, which is summarized in Algorithm 2, provides Type I error control for $H_{0,j,S}^{\text{LC}}$, as the following proposition summarizes (the proof is included in Appendix C.3).

**Proposition 2.** $\forall t \geq 1$, $\kappa_t := \tanh(\rho_t^{\text{X-SKIT}}(\hat{Y}_{S\cup\{j\}}) - \rho_t^{\text{X-SKIT}}(\hat{Y}_S))$ provides a fair game for $H_{0,j,S}^{LC}$.

**Computational complexity of X-SKIT.** Note that Algorithm 2 assumes access to a sampler $P_{H|Z_C=z_C}$, so its computational complexity is $\mathcal{O}(T_n \tau^2)$, where, similarly to above, $T_n$ is the cost of the sampler. We briefly remark that, for the nonparametric samplers used in this work, $T_n = n^2$ (compared to $\tau n^2$ for C-SKIT) because we only need to estimate one conditional distribution.

So far, we have presented our tests for *one concept at a time*, but we are interested in testing $m \geq 1$ concepts. In this setting, it is well-known that multiple hypothesis testing requires appropriate corrections to avoid inflated significance levels. We use a result of Wang and Ramdas [79] and devise a greedy post-processor that guarantees false discovery rate control [3] (see Appendix A.4).

## 4 Results

First, we verify that our tests are valid and that they are adaptive to the hardness of their null hypotheses on two synthetic experiments in Appendix D. Here, we showcase the flexibility and effectiveness of our framework on zero-short image classification across several VL models on three

Table 1: Summary of results for each dataset. Metrics are reported as average across all VL models used in the experiments. See main text for details about the models and the metrics used.

| Method | Original model | Imagenette | | AwA2 | | | CUB | | |
|---|---|---|---|---|---|---|---|---|---|
| | | Accuracy | Rank agreement | Accuracy | Rank agreement | $f_1$ | Accuracy | Rank agreement | $f_1$ |
| SKIT | ✓ | | 0.51 | | **0.50** | **0.65** | | 0.82 | 0.93 |
| C-SKIT | ✓ | **98.99**% | 0.54 | **99.50**% | 0.46 | 0.57 | **89.52**% | - | - |
| X-SKIT | ✓ | | **0.59** | | - | - | | - | - |
| PCBM | ✗ | 95.85% | 0.45 | 95.11% | 0.36 | 0.53 | - | - | - |

(a) Global importance with SKIT.  (b) Global conditional importance with C-SKIT.

Figure 2: Importance results with CLIP:ViT-L/14 on 2 classes in the AwA2 dataset. Concepts are annotated with *(p)* if they are present in the class, or with *(a)* otherwise.

real-world datasets: Animal with Attributes 2 (AwA2) [82], CUB-200-2011 (CUB) [77], and the Imagenette subset of ImageNet [22].[3] We compare performance and transferability of the ranks of importance provided by each method across 8 VL models (see Appendix E for details) and, for all experiments, $f$ is the image encoder of the model and $g$ is the (linear) zero-shot classifier constructed by encoding *"A photo of a <CLASS_NAME>"* with the text encoder. Herein, we will always use RBF kernels to compute payoffs, and we repeat each test 100 times on independent draws of $\tau^{\max}$ samples to estimate each concept's *expected rejection time* and *expected rejection rate* at a significance level of $\alpha = 0.05$ with the FDR post-processor described in Appendix A.4. That is, a (normalized) rejection time of 1 means failing to reject in $\tau^{\max}$ steps. Finally, recall that C-SKIT and X-SKIT need access to $P_{Z_j|Z_{-j}}$ and $P_{H|Z_C=z_C}$, and that these distributions are not known in general. Since $m$ is small, we use nonparametric methods to estimate them (see Appendix E.1).

Table 1 summarizes the results of all experiments, which we now present and discuss individually.

## 4.1 AwA2 Dataset

Given the presence of global (i.e., class-level) annotations, we use SKIT and C-SKIT to test the global (and global conditional) semantic importance structure of the predictions for the top-10 best classified animal categories across all models (we describe the dataset, the concepts used, and the hyperparameters of the tests in Appendix E.2). We classify the top-10 concepts reported by each method as important, and we compute the $f_1$ score with the ground-truth annotations. We briefly remark that this choice is informed by the fact that most concepts have rejection rates larger than the significance level of $\alpha$. When comparing with PCBM—since we use different concepts for each class—we train 100 independent linear models for each class, and we rank concepts based on their average absolute weights (instead of signed ones) because the null hypotheses presented in this work are two-sided, i.e. a concept is important both if it increases the prediction for a class or if it decreases it. Table 1 shows that both SKIT and C-SKIT outperform PCBM across all three metrics, with SKIT providing the best average rank agreement across different models and importance $f_1$ score (0.50 and 0.65, respectively). The fact that ranks provided by our tests have higher average agreement compared to PCBM suggests that VL models may share a similar semantic independence structure notwithstanding their embedding size or training strategy, i.e. semantic importance may be *transferable* across models (all individual pairwise agreements are included in Fig. E.4).

Finally, Fig. 2 shows ranks of importance with CLIP:ViT-L/14 on 2 animal categories (see Figs. E.5 and E.6 for all classes). In general, concepts are globally important (rejection rates are greater than $\alpha$),

---

[3]Code to reproduce all experiments is available at https://github.com/Sulam-Group/IBYDMT.

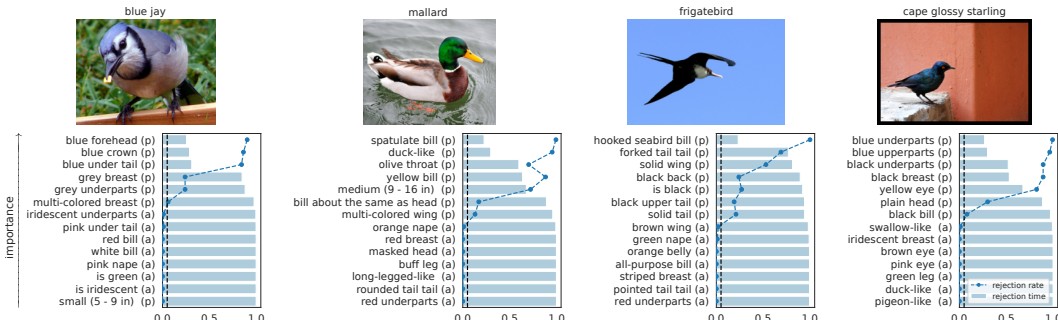

Figure 3: Importance results with X-SKIT and CLIP:ViT-L/14 on 4 images in the CUB dataset. Concepts with *(p)* are present in the image according to human annotations, and *(a)* otherwise.

and it is harder to reject the global conditional null hypothesis (rejection rates are lower and rejection times larger), naturally reflecting the fact that conditional independence is a stronger condition.

## 4.2 CUB Dataset

This dataset (differently from AwA2) provides per-image annotations of semantic attributes. So, we use X-SKIT to test the semantic importance structure of VL models locally on particular images and validate its performance against such annotations (we include details about this experiment and extended results in Appendix E.3). The purpose of this experiment is to validate the performance of X-SKIT, hence we use the ground-truth binary semantic annotations as an oracle instead of predicting the presence of concepts. In practical scenarios where ground-truth is not available, one could—as done by previous works [12]—use LLMs to answer binary questions (e.g., "Does this bird have an orange bill? Yes/No"). Furthermore, note that for each concept $j \in [m]$ there are exponentially many tests with null hypothesis $H_{0,j,S}^{\text{LC}}$—one for each subset $S \subseteq [m] \setminus \{j\}$—which are intractable to compute. Thus, we report average results over 100 tests with random subsets with fixed size $s$.

Fig. 3 depicts prototypical results with CLIP:ViT-L/14 (Fig. E.9 includes results for all models on the same images). After running X-SKIT, we classify concepts as important by thresholding their rejection rates at level $\alpha$—which is a statistically-valid way of selecting important concepts. Results are included in Table 1, and we conclude that X-SKIT provides ranks of importance that are well-aligned both across models (0.82 rank agreement) and with ground-truth annotations ($f_1$ score of 0.93). We remark that X-SKIT is the first method to provide local semantic explanations, hence why we cannot compare with alternatives.

## 4.3 Imagenette Dataset

Lastly, we use both SKIT, C-SKIT, and X-SKIT on the Imagenette subset of ImageNet [22]—which does not provide ground-truth semantic annotations to evaluate performance with. So, we use SpLiCe [6] to select which concepts to test (see Appendix E.4 for details), but we stress that any user-defined set of concepts would be valid—a unique feature of our proposed framework.

Figs. 4a and 4b show SKIT and C-SKIT results with CLIP:ViT-L/14 on 2 classes in the dataset (Fig. E.11 includes all classes). We use SpLiCe to encode the entire dataset and test the top-20 concepts. Analogous to the experiment on AwA2, we can see that rejection rates are lower for C-SKIT (i.e., conditional dependence) compared to SKIT (i.e., marginal dependence). We evaluate rank agreement across all models and compare with PCBM in Table 1. These results confirm that not only are the ranks produced by our tests more transferable across models (rank agreement of 0.51 for SKIT, 0.54 or C-SKIT, 0.59 for X-SKIT, and 0.45 for PCBM), but also they retain the performance of the original classifier (98.99% for our methods vs 95.55% for PCBM). We refer interested readers to Fig. E.12 for all pairwise comparisons. Furthermore, we qualitatively study the stability of our tests as a function of $\tau^{\max}$ in Fig. E.13. This is important because $\tau^{\max}$ represents the sample complexity of the tests. Our findings indicate that important concepts tend to exhibit greater stability in their ranks compared to less important ones, with SKIT showing overall more stability than C-SKIT.

To conclude, Fig. 4c shows X-SKIT results with CLIP:ViT-L/14 on three random images from the dataset (see Figs. E.15 and E.16 for all models and more images). We use SpLiCe to encode each image and obtain its top-10 concepts, and then add the bottom-4 according to PCBM, for a total of 14 attributes per image. The choice of combining concepts both from SpLiCe and PCBM will highlight

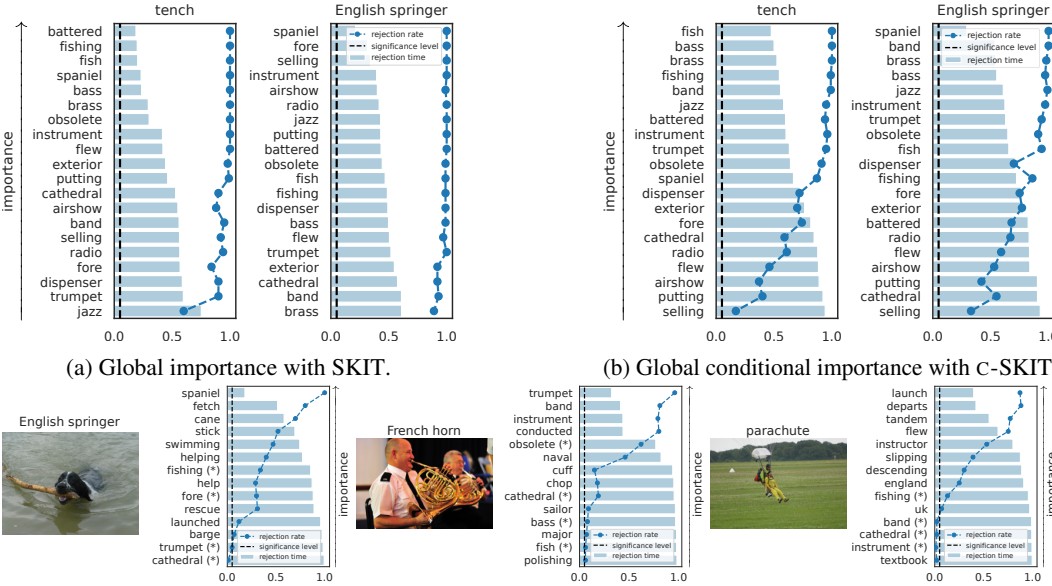

(a) Global importance with SKIT.

(b) Global conditional importance with C-SKIT.

(c) Local importance with X-SKIT. The bottom-4 concepts according to PCBM are annotated with *(*)*.

Figure 4: Results with CLIP:ViT-L/14 on Imagenette.

the differences between these methods and our notion of local statistical importance, as we will shortly expand on. Recall that we use nonparametric samplers to approximate $P_{H|Z_C=z_C}$, so the cost of using image-specific concepts boils down to projecting the feature embeddings with a different matrix $c$, which is negligible compared to running the tests. We note that parametric generative models—such as variational autoencoders or diffusion models—would have required retraining for each set of concepts, which is expensive. Overall, we find that ranks are well-aligned across models (0.71 rank agreement, see Table 1). We can appreciate how the bottom-4 concepts from PCBM, which are annotated with an asterisk, are not always last, i.e. a concept may be locally important even if it is not globally important. For example, concept "fishing" may not be globally important for class "English springer", but it is locally important for an image of a dog in water. Conversely, a concept having a high weight according to SpLiCe does not imply it will be statistically important for the predictions of the model, and these distinctions are important in order to communicate findings transparently.

## 5 Conclusions

There exist an increasing interest in explaining modern, unintelligible predictors and, in particular, doing so with inherently interpretable concepts that convey specific meaning to users. This work is the first to formalize precise statistical notions of semantic importance in terms of global (i.e., over a population) and local (i.e., on a sample) conditional hypothesis testing. We propose novel, valid tests for each notion of importance while providing a rank of importance by deploying ideas of sequential testing. Importantly, by approaching importance via conditional independence (and by developing appropriate valid tests), we are able to provide Type I error and FDR control, a feature that is unique to our framework compared to existing alternatives. Furthermore, our tests allow to explain the original—potentially nonlinear—classifier that would be used in the wild, as opposed to training surrogate linear models as has been the standard so far.

Naturally, our work has limitations. First and foremost, the procedures introduced in this work require access to samplers, and there might be settings were learning these models is hard; we used nonparametric estimators in our experiments, but modern generative models could be employed, too. Second, kernel-based tests rely on the assumption that the kernels used are characteristic for the space of distributions considered. Although these assumptions are usually satisfied in $\mathbb{R}^d$ for RBF kernels, there may exist data modalities where this is not true (e.g., discrete data, graphs), which would compromise the power of the test. Finally, although we grant full flexibility to users to specify the concepts they care about, there is no guarantee that these are well-represented in the feature space of the model, nor that they are the most informative ones for a specific task. All these points are a matter of ongoing and future work.

## Acknowledgments

We sincerely thank Zhenzhen Wang and the anonymous NeurIPS reviewers for useful conversations that strengthened the presentation of our experimental results. This research was supported by NSF CAREER Award CCF 2239787.

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

# A  Testing by Betting

In this appendix, we include additional background information on testing by betting that was omitted from the main text for the sake of conciseness of presentation. Recall that the wealth process $\{K_t\}_{t \in \mathbb{N}_0}$ with $\mathbb{N}_0 := \mathbb{N} \cup \{0\}$ is defined as

$$K_0 = 1 \quad \text{and} \quad K_t = K_{t-1} \cdot (1 + v_t \kappa_t) \tag{6}$$

where $v_t \in [-1, 1]$ is the betting fraction and $\kappa_t \in [-1, 1]$ the payoff of the bet.

## A.1  Test Martingales

We start by introducing the definition of a *test martingale* (see, for example, Shaer et al. [58]).

**Definition A.1** (Test martingale). A nonnegative stochastic process $\{S_t\}_{t \in \mathbb{N}_0}$ is a test martingale if $S_0 = 1$ and, under a null hypothesis $H_0$, it is a supermartingale, i.e.

$$\mathbb{E}_{H_0}[S_t \mid \mathcal{F}_{t-1}] \leq S_{t-1}, \tag{7}$$

where $\mathcal{F}_{t-1}$ is the filtration (i.e., the smallest $\sigma$-algebra) of all previous observations.

In the following, we will use Ville's inequality, which we include for the sake of completeness.

**Lemma A.1** (Ville's inequality [75]). *If the stochastic process $\{S_t\}_{t \in \mathbb{N}_0}$ is a nonnegative supermartingale,*

$$\mathbb{P}[\exists t \geq 0 : \ S_t \geq \eta] \leq \mathbb{E}[S_0]/\eta, \ \forall \eta > 0. \tag{8}$$

With this, we state a condition under which we can use the wealth process to reject a null hypothesis $H_0$ with Type I error control.

**Lemma A.2** (See Shaer et al. [58], Shekhar and Ramdas [62]). *If*

$$\mathbb{E}_{H_0}[\kappa_t \mid \mathcal{F}_{t-1}] = 0, \tag{9}$$

*where $\mathcal{F}_{t-1}$ denotes the filtration (i.e., the smallest $\sigma$-algebra) of all previous observations, then the wealth process $\{K_t\}_{t \in \mathbb{N}_0}$ describes a fair game and*

$$\mathbb{P}_{H_0}[\exists t \geq 1 : \ K_t \geq 1/\alpha] \leq \alpha. \tag{10}$$

*Proof.* It suffices to show that if $\mathbb{E}_{H_0}[\kappa_t \mid \mathcal{F}_{t-1}] = 0$, then the wealth process $\{K_t\}_{t \in \mathbb{N}_0}$ is a test martingale:

1. $K_0 = 1$ by definition, and

2. It is immediate to see that the wealth process is nonnegative because $v_t \kappa_t \in [-1, 1]$ and the bettor never risks more than their current wealth, i.e. they will never go into debt. Finally,

3. If $\mathbb{E}_{H_0}[\kappa_t \mid \mathcal{F}_{t-1}] = 0$, then

$$\mathbb{E}_{H_0}[K_t \mid \mathcal{F}_{t-1}] = \mathbb{E}_{H_0}[K_{t-1} \cdot (1 + v_t \kappa_t) \mid \mathcal{F}_{t-1}] \tag{11}$$
$$= K_{t-1} \cdot \mathbb{E}_{H_0}[1 + v_t \kappa_t \mid \mathcal{F}_{t-1}] \quad (K_{t-1} \mid \mathcal{F}_{t-1} \text{ is constant}) \tag{12}$$
$$\leq K_{t-1} \cdot (1 + \mathbb{E}_{H_0}[\kappa_t \mid \mathcal{F}_{t-1}]) \quad (v_t \leq 1) \tag{13}$$
$$= K_{t-1}, \tag{14}$$

and the wealth process is a supermartingale under the null.

Then, by Ville's inequality, we conclude that for any significance level $\alpha \in (0, 1)$

$$\mathbb{P}_{H_0}[\exists t \geq 1 : \ K_t \geq 1/\alpha] \leq \alpha \mathbb{E}[K_0] = \alpha \tag{15}$$

which is the statement of the lemma. $\square$

## A.2 Symmetry-based Two-sample Sequential Testing

In this section, we show how to leverage symmetry to construct valid sequential tests for a two-sample null hypothesis of the form

$$H_0 : P = \widetilde{P}. \tag{16}$$

**Lemma A.3** (See [51, 58, 62]). *$\forall t \geq 1$, let $X \sim P$ and $\widetilde{X} \sim \widetilde{P}$ be two random variables sampled from $P$ and $\widetilde{P}$, respectively. If $P = \widetilde{P}$, it holds that for any fixed function $\rho_t : \mathcal{X} \to \mathbb{R}$*

$$\rho_t(X) - \rho_t(\widetilde{X}) \stackrel{d}{=} \rho_t(\widetilde{X}) - \rho_t(X), \tag{17}$$

*that is*

$$p_0(\rho_t(X) - \rho_t(\widetilde{X})) = p_0(\rho_t(\widetilde{X}) - \rho_t(X)), \tag{18}$$

*where $p_0$ is the probability density function induced by $H_0$.*

*Proof.* The proof is straightforward. If $P = \widetilde{P}$, then $X$ and $\widetilde{X}$ are exchangeable by assumption. □

### Proof of Lemma 2

Recall the lemma states that for any fixed function $\rho_t : \mathcal{X} \to \mathbb{R}$, the payoff

$$\kappa_t = \tanh(\rho_t(X) - \rho_t(\widetilde{X})) \tag{19}$$

provides a fair game (i.e., it satisfies Lemma A.2) for a two-sample test with null hypothesis $H_0 : P = \widetilde{P}$. We use Lemma A.3 above to prove a stronger result that implies the desired claim.

**Lemma A.4** (See [51, 58, 62]). *For any $t \geq 1$, and any fixed anti-symmetric function $\xi : \mathbb{R} \to \mathbb{R}$, it holds that*

$$\mathbb{E}_{H_0}[\xi(\rho_t(X) - \rho_t(\widetilde{X})) \mid \mathcal{F}_{t-1}] = 0. \tag{20}$$

*Proof.* We can see that

$$\mathbb{E}_{H_0}[\xi(\rho_t(X) - \rho_t(\widetilde{X})) \mid \mathcal{F}_{t-1}] = \mathbb{E}_{H_0}[\xi(\rho_t(X) - \rho_t(\widetilde{X}))] \qquad (\rho_t, \xi \text{ are fixed}) \tag{21}$$

$$= \int_{\mathbb{R}} \xi(u)p_0(u) \, \mathrm{d}u \qquad (\text{change of variables}) \tag{22}$$

$$= \int_{\mathbb{R}_+} (\xi(u) + \xi(-u))p_0(u) \, \mathrm{d}u \qquad (\text{by Lemma A.3}) \tag{23}$$

$$= \int_{\mathbb{R}_+} (\xi(u) - \xi(u))p_0(u) \, \mathrm{d}u \qquad (\xi \text{ is anti-symmetric}) \tag{24}$$

$$= 0, \tag{25}$$

which concludes the proof. □

*Proof of Lemma 2.* Note that $\tanh$ is an anti-symmetric function, so Lemma A.4 holds. Then, Lemma A.2 implies that $\kappa_t = \tanh(\rho_t(X) - \rho_t(\widetilde{X}))$ provides a test martingale for $H_0 : P = \widetilde{P}$. □

## A.3 Betting Strategies

So far, we have discussed how to construct valid test martingales in terms of the payoff $\kappa_t$. Then, it remains to define a strategy to choose the betting fraction $v_t$. In general, any method that picks $v_t$ *before* data is revealed maintains validity of the test, and we briefly summarize a few alternatives.

**Constant betting fraction.** Naturally, a fixed betting fraction $v_t = v$ is valid. However, this strategy may be prone to *overshooting*, i.e. the wealth may go to zero almost surely under the alternative, and severely impact the power of the test [51, Example 2].

**Mixture method [17, 58].** A possible way to overcome the limitations of setting a fixed betting fraction is to average across a distribution, i.e.

$$K_t = \int_{\mathcal{V}} K_t^{(v)} p(v) \, \mathrm{d}v, \tag{26}$$

where $K_t^{(v)}$ is the wealth with constant betting fraction $v_t = v$, and $p(v)$ is a prior over the choice of fractions (e.g., uniform over $[-1, 1]$). This choice is valid, and motivated by the intuition that the mixture martingale will be driven by the term that achieves the optimal betting fraction [58, Theorem 1].

**Online Newton step (ONS) [19].** Alternatively, one can frame choosing the betting fraction as an online optimization problem that finds the optimal $v_t$ in terms of the regret of the strategy. We refer interested readers to [19, 58, 62] for a theoretical analysis of this strategy and simply state here the wealth's growth rate. Algorithm A.1 summarizes this strategy.

**Lemma A.5** (See Shekhar and Ramdas [62]). *For any sequence $\{v_t \in [-1, 1] : t \geq 1\}$, it holds that*

$$\log K_t \geq \frac{1}{8t} \left( \sum_{t'=1}^{t} v_{t'} \right)^2 - \log t. \tag{27}$$

---

**Algorithm A.1** ONS Betting Strategy

---

**Input:** Sequence of payoffs $\{\kappa_t\}_{t \geq 1}$

1: $a_0 \leftarrow 1$
2: $v_1 \leftarrow 0$
3: **for** $t \geq 1$ **do**
4:     $z_t \leftarrow \kappa_t/(1 + v_t \kappa_t)$
5:     $a_t \leftarrow a_{t-1} + z_t^2$
6:     $v_{t+1} \leftarrow \max(0, \min(1, a_t + 2/(2 - \log(3)) \cdot z_t/a_t))$
7: **end for**

---

## A.4 Controlling False Discovery Rate

Finally, we briefly present one way to provide false discovery rate (FDR) control when testing multiple hypotheses with sequential tests. Given $m$ null hypotheses $H_0^{(1)}, \ldots, H_0^{(m)}$, denote $e^{(1)}, \ldots, e^{(m)}$ their respective $e$-values [60, 76] and let $\mathcal{E} : [0, \infty]^m \to 2^{[m]}$ be an $e$-testing procedure such that $\widetilde{S} = \mathcal{E}(e^{(1)}, \ldots, e^{(m)})$ is the set of rejected null hypotheses. Then, FDR is the expected proportion of false discoveries to the number of total findings, i.e.

$$\text{FDR} := \mathbb{E} \left[ \frac{|\widetilde{S} \cap S_0|}{|\widetilde{S}|} \right], \tag{28}$$

where $S_0 := \{j \in [m] : H_0^{(j)} \text{ is true}\}$ is the set of true null hypotheses (i.e., the ones that should not be rejected). Following [8, 79], we say that $\mathcal{E}$ is *self-consistent at level* $\alpha$ if every rejected $e$-value satisfies $e^{(j)} \geq m/\alpha|\widetilde{S}|$, and we now state the lemma we use to construct our FDR post-processor in Algorithm A.2.

**Lemma A.6** (See Wang and Ramdas [79]). *Any self-consistent $e$-testing procedure at level $\alpha$ controls FDR at level $\alpha$ for arbitrary configurations of $e$-values.*

---

**Algorithm A.2** Level-$\alpha$ greedy FDR post-processor.

---

**Input:** Wealth processes $\{K_t^{(1)}\}, \ldots, \{K_t^{(m)}\}, t \in \mathbb{N}_0$.

1: $\widetilde{S} \leftarrow \emptyset$
2: **for** $s = 1, \ldots, m$ **do**
3:     $j', \tau' \leftarrow \underset{j \in [m] \setminus \widetilde{S},\ \tau \in [0, \infty]}{\arg \min} \tau \text{ s.t. } K_\tau^{(j)} \geq m/\alpha s$
4:     **if** $\tau' = \infty$ **then**
5:         **return** $\widetilde{S}$
6:     **end if**
7:     $\widetilde{S} \leftarrow \widetilde{S} \cup \{j'\}$
8: **end for**
9: **return** $\widetilde{S}$

---

Recall that the optional stopping theorem implies that for a test martingale $\{K_t\}_{t \in \mathbb{N}_0}$, the wealth $K_t$ is an $e$-value for any $t \geq 1$. Then, intuitively, Algorithm A.2 transforms an $e$-testing procedure $\mathcal{E}$ into a self-consistent one by greedily rejecting concepts as soon as they cross the adjusted threshold $m/\alpha|\widetilde{S}|$. Note that we do not know the number of rejections *a priori*, and that $m/\alpha|\widetilde{S}|$ is a decreasing function of $|\widetilde{S}|$. Hence, the adjusted threshold for the first concept will be $m/\alpha$ (which matches the common Bonferroni correction [9]), $m/2\alpha$ for the second one, then $m/3\alpha$, and so on and so forth. The procedure stops when no more concepts reach the threshold, and concepts are sorted by their adjusted rejection times. We remark that Algorithm A.2 runs in $\mathcal{O}(m)$ time, and that it does not change the individual testing procedures—which is important because concepts are tested concurrently in practice.

# B  Technical Details on Payoff Functions

In this appendix, we include technical details on how to compute the payoff functions of all tests presented in this paper. We start with a brief overview of the maximum mean discrepancy (MMD) [29], and we refer interested readers to [1, 4, 61, 68] for rigorous introductions to the theory of reproducing kernel Hilbert spaces (RKHSs) and their applications to probability and statistics.

**Definition B.1** (Mean embedding (see Gretton et al. [29]))**.** Let $P$ be a probability distribution on $\mathcal{X}$ and $\mathcal{R}$ an RKHS on the same domain. The mean embedding of $P$ in $\mathcal{R}$ is the element $\mu_P \in \mathcal{R}$ with

$$\forall \rho \in \mathcal{R}, \ \mathbb{E}_P[\rho(X)] = \langle \mu_P, \rho \rangle_{\mathcal{R}}. \tag{29}$$

Furthermore, given $X^{(1)}, \ldots, X^{(n)}$ sampled i.i.d. from $P$, the plug-in estimate $\hat{\mu}_P^{(n)}$ is

$$\hat{\mu}_P^{(n)} := \frac{1}{n} \sum_{i=1}^{n} \varphi(X^{(i)}), \tag{30}$$

where $\varphi$ is the canonical feature map, i.e. $\varphi(X) = k(X, \cdot)$, and $k$ is the kernel associated with $\mathcal{R}$.

We now define the MMD between two probability distributions $P, Q$ and show that it can be rewritten in terms of their mean embeddings.

**Definition B.2** (Integral probability metric (see Müller [45]))**.** Let $P, Q$ be two probability distributions over $\mathcal{X}$. Furthermore, denote $\mathcal{G} = \{g : \mathcal{X} \to \mathbb{R}\}$ a hypothesis class of real-valued functions over $\mathcal{X}$. Then,

$$D_{\mathcal{G}}(P, Q) := \sup_{g \in \mathcal{G}} |\mathbb{E}_P[g(X)] - \mathbb{E}_Q[g(X)]| \tag{31}$$

is the distance between $P$ and $Q$ induced by $\mathcal{G}$, and the function $g^*$ that achieves the supremum is called *witness function*.

The MMD is defined as $D_{\mathcal{B}(\mathcal{R})}(P, Q)$, where $\mathcal{B}(\mathcal{R})$ is the unit ball of $\mathcal{R}$, i.e.

$$\mathcal{B}(\mathcal{R}) := \{\rho \in \mathcal{R} : \ \|\rho\|_{\mathcal{R}} \leq 1\}. \tag{32}$$

**Definition B.3** (Maximum mean discrepancy (see Gretton et al. [29]))**.** For $P, Q$ defined as above, let $\mathcal{R}$ be an RKHS on their domain. Then,

$$\text{MMD}(P, Q) := \sup_{\rho \in \mathcal{B}(\mathcal{R})} \mathbb{E}_P[\rho(X)] - \mathbb{E}_Q[\rho(X)]. \tag{33}$$

We note that we drop the absolute value because if $\rho \in \mathcal{B}(\mathcal{R})$, then $-\rho \in \mathcal{B}(\mathcal{R})$ also. From the definition of mean embedding, it follows that

$$\text{MMD}(P, Q) = \sup_{\rho \in \mathcal{B}(\mathcal{R})} \langle \mu_P, \rho \rangle_{\mathcal{R}} - \langle \mu_Q, \rho \rangle_{\mathcal{R}} \tag{34}$$

$$= \sup_{\rho \in \mathcal{B}(\mathcal{R})} \langle \mu_P - \mu_Q, \rho \rangle \tag{35}$$

$$= \|\mu_P - \mu_Q\|_{\mathcal{R}}, \tag{36}$$

and its witness function satisfies

$$\rho^* \propto \mu_P - \mu_Q. \tag{37}$$

**Algorithm B.1** Level-$\alpha$ SKIT for the global importance of concept $j$

---

**Input:** Stream $(\hat{Y}^{(t)}, Z_j^{(t)}) \sim P_{\hat{Y}Z_j}$.

1: $K_0 \leftarrow 1$
2: Initialize ONS strategy as in Algorithm A.1.
3: **for** $t = 1, \ldots$ **do**
4:      Compute $\rho_t$ as in Eq. (47)
5:      Observe $D^{(2t-1)} = (\hat{Y}^{(2t-1)}, Z_j^{(2t-1)})$    and    $D^{(2t)} = (\hat{Y}^{(2t)}, Z_j^{(2t)})$
6:      $\widetilde{D}^{(2t-1)} \leftarrow (\hat{Y}^{(2t-1)}, Z_j^{(2t)})$    and    $\widetilde{D}^{(2t)} \leftarrow (\hat{Y}^{(2t)}, Z_j^{(2t-1)})$
7:      Compute $\kappa_t$ as in Eq. (48)
8:      $K_t \leftarrow K_{t-1} \cdot (1 + v_t \kappa_t)$
9:      **if** $K_t \geq 1/\alpha$ **then**
10:         **return** $t$
11:      **end if**
12:      $v_{t+1} \leftarrow$ ONS step
13: **end for**

---

## B.1 Computing $\rho_t^{\text{SKIT}}$ and $\kappa_t^{\text{SKIT}}$

Recall that $\rho_t^{\text{SKIT}}$ is the estimate of the witness function of $\text{MMD}(P_{\hat{Y}Z_j}, P_{\hat{Y}} \times P_{Z_j})$ at time $t$, i.e.

$$\rho_t^{\text{SKIT}} = \hat{\mu}_{\hat{Y}Z_j}^{(2(t-1))} - \hat{\mu}_{\hat{Y}}^{(2(t-1))} \otimes \hat{\mu}_{Z_j}^{(2(t-1))}, \tag{38}$$

where

$$\hat{\mu}_{\hat{Y}Z_j}^{(2(t-1))} = \frac{1}{2(t-1)} \sum_{t'=0}^{2(t-1)} (\varphi_{\hat{Y}}(\hat{Y}^{(t')}) \otimes \varphi_{Z_j}(Z_j^{(t')})), \tag{39}$$

$$\hat{\mu}_{\hat{Y}}^{(2(t-1))} = \frac{1}{t-1} \sum_{t'=0}^{2(t-1)} \varphi_{\hat{Y}}(\hat{Y}^{(t')}), \qquad \hat{\mu}_{Z_j}^{(2(t-1))} = \frac{1}{t-1} \sum_{t'=0}^{2(t-1)} \varphi_{Z_j}(Z_j^{(t')}), \tag{40}$$

and $\varphi_{\hat{Y}}, \varphi_{Z_j}$ are the canonical feature maps associated with $\mathcal{R}_{\hat{Y}}$ and $\mathcal{R}_{Z_j}$, respectively. We remark that $\rho_t^{\text{SKIT}}$ is an operator, and, for a sample $(\hat{y}, z_j)$, its value $\rho_t^{\text{SKIT}}(\hat{y}, z_j)$ can be computed as

$$\rho_t^{\text{SKIT}}(\hat{y}, z_j) = (\hat{\mu}_{\hat{Y}Z_j}^{(2(t-1))} - \hat{\mu}_{\hat{Y}}^{(2(t-1))} \otimes \hat{\mu}_{Z_j}^{(2(t-1))})(\hat{y}, z_j) \tag{41}$$

$$= \hat{\mu}_{\hat{Y}Z_j}^{(2(t-1))}(\hat{y}, z_j) - (\hat{\mu}_{\hat{Y}}^{(2(t-1))} \otimes \hat{\mu}_{Z_j}^{(2(t-1))})(\hat{y}, z_j) \tag{42}$$

with

$$\hat{\mu}_{\hat{Y}Z_j}^{(2(t-1))}(\hat{y}, z_j) = \frac{1}{2(t-1)} \sum_{t'=0}^{2(t-1)} k_{\hat{Y}}(\hat{Y}^{(t')}, \hat{y}) k_{Z_j}(Z_j^{(t')}, z_j) \tag{43}$$

$$(\hat{\mu}_{\hat{Y}}^{(2(t-1))} \otimes \hat{\mu}_{Z_j}^{(2(t-1))})(\hat{y}, z_j) = \frac{1}{2(t-1)} \left[ \sum_{t'=0}^{2(t-1)} k_{\hat{Y}}(\hat{Y}^{(t')}, \hat{y}) \cdot \sum_{t'=0}^{2(t-1)} k_{Z_j}(Z_j^{(t')}, z_j) \right], \tag{44}$$

where $k_{\hat{Y}}, k_{Z_j}$ are the kernels associated with $\mathcal{R}_{\hat{Y}}$ and $\mathcal{R}_{Z_j}$, respectively.

Furthermore, note that, in practice, we only have access to samples from the test distribution $P_{\hat{Y}Z_j}$ (i.e., the joint) and we swap elements of two consecutive samples to simulate data from the null distribution $P_{\hat{Y}} \times P_{Z_j}$. More formally, let

$$D^{(2t)} = (\hat{Y}^{(2t)}, Z_j^{(2t)}) \sim P_{\hat{Y}Z_j}, \qquad D^{(2t-1)} = (\hat{Y}^{(2t-1)}, Z_j^{(2t-1)}) \sim P_{\hat{Y}Z_j} \tag{45}$$

such that

$$\widetilde{D}^{(2t)} = (\hat{Y}^{(2t)}, Z_j^{(2t-1)}), \qquad \widetilde{D}^{(2t-1)} = (\hat{Y}^{(2t-1)}, Z_j^{(2t)}). \tag{46}$$

Then,

$$\rho_t^{\text{SKIT}} := \hat{\mu}_{\hat{Y}Z_j}^{(2(t-1))} - \hat{\mu}_{\hat{Y}}^{(2(t-1))} \otimes \hat{\mu}_{Z_j}^{(2(t-1))} \tag{47}$$

where $\hat{\mu}_{\hat{Y}Z_j}, \hat{\mu}_{\hat{Y}}, \hat{\mu}_{Z_j}$ are the *mean embeddings* of $P_{\hat{Y}Z_j}, P_{\hat{Y}}, P_{Z_j}$ in $\mathcal{R}_{\hat{Y}} \otimes \mathcal{R}_{Z_j}, \mathcal{R}_{\hat{Y}}$, and $\mathcal{R}_{Z_j}$, respectively, and $\otimes$ is the tensor product as in Gretton et al. [29]. We remark that $\rho_t^{\text{SKIT}}$ is a real-valued operator, i.e. $\rho_t^{\text{SKIT}} : \mathbb{R} \times \mathbb{R} \to \mathbb{R}$, and that we use data up to $t-1$ to compute $\rho_t$ in order to maintain validity of the test, i.e. $\rho_t$ is fixed conditionally on previous observations.

Following Lemma 2, we conclude

$$\kappa_t^{\text{SKIT}} := \tanh\left(\rho_t^{\text{SKIT}}(D^{(2t-1)}) + \rho_t^{\text{SKIT}}(D^{(2t)}) - \rho_t^{\text{SKIT}}(\widetilde{D}^{(2t-1)}) - \rho_t^{\text{SKIT}}(\widetilde{D}^{(2t)})\right) \tag{48}$$

and Algorithm B.1 summarizes the SKIT procedure for the global semantic importance null hypothesis $H_{0,j}^{\text{G}}$ in Definition 1.

## B.2   Computing $\rho_t^{\text{C-SKIT}}$ and $\kappa_t^{\text{C-SKIT}}$

Recall that $\rho_t^{\text{C-SKIT}}$ is the estimate of the witness function of $\text{MMD}(P_{\hat{Y}Z_jZ_{-j}}, P_{\hat{Y}\widetilde{Z}_jZ_{-j}})$ with $\widetilde{Z}_j \sim P_{Z_j|Z_{-j}}$ at time $t$, i.e.

$$\rho_t^{\text{C-SKIT}} = \hat{\mu}_{\hat{Y}Z_jZ_{-j}}^{(t-1)} - \hat{\mu}_{\hat{Y}\widetilde{Z}_jZ_{-j}}^{(t-1)}, \tag{49}$$

where

$$\hat{\mu}_{\hat{Y}Z_jZ_{-j}}^{(t-1)} = \frac{1}{t-1} \sum_{t'=0}^{t-1} \left( \varphi_{\hat{Y}}(\hat{Y}^{(t')}) \otimes \varphi_{Z_j}(Z_j^{(t')}) \otimes \varphi_{Z_{-j}}(Z_{-j}^{(t')}) \right) \tag{50}$$

$$\hat{\mu}_{\hat{Y}\widetilde{Z}_jZ_{-j}}^{(t-1)} = \frac{1}{t-1} \sum_{t'=0}^{t-1} \left( \varphi_{\hat{Y}}(\hat{Y}^{(t')}) \otimes \varphi_{Z_j}(\widetilde{Z}_j^{(t')}) \otimes \varphi_{Z_{-j}}(Z_{-j}^{(t')}) \right) \tag{51}$$

and $\varphi_{\hat{Y}}, \varphi_{Z_j}, \varphi_{Z_{-j}}$ are the canonical feature maps associated with their respective RKHSs. We remark that $\rho_t^{\text{C-SKIT}}$ is defined as an operator, and, for a triplet $(\hat{y}, z_j, z_{-j})$ its value can be computed as

$$\rho_t^{\text{C-SKIT}}(\hat{y}, z_j, z_{-j}) = (\hat{\mu}_{\hat{Y}Z_jZ_{-j}}^{(t-1)} - \hat{\mu}_{\hat{Y}\widetilde{Z}_jZ_{-j}}^{(t-1)})(\hat{y}, z_j, z_{-j}) \tag{52}$$

$$= \hat{\mu}_{\hat{Y}Z_jZ_{-j}}^{(t-1)}(\hat{y}, z_j, z_{-j}) - \hat{\mu}_{\hat{Y}\widetilde{Z}_jZ_{-j}}^{(t-1)}(\hat{y}, z_j, z_{-j}) \tag{53}$$

with

$$\hat{\mu}_{\hat{Y}Z_jZ_{-j}}^{(t-1)}(\hat{y}, z_j, z_{-j}) = \frac{1}{t-1} \sum_{t'=0}^{t-1} k_{\hat{Y}}(\hat{Y}^{(t')}, y) k_{Z_j}(Z_j^{(t')}, z_j) k_{Z_{-j}}(Z_{-j}^{(t')}, z_{-j}) \tag{54}$$

$$\hat{\mu}_{\hat{Y}\widetilde{Z}_jZ_{-j}}^{(t-1)}(\hat{y}, z_j, z_{-j}) = \frac{1}{t-1} \sum_{t'=0}^{t-1} k_{\hat{Y}}(\hat{Y}^{(t')}, y) k_{Z_j}(\widetilde{Z}_j^{(t')}, z_j) k_{Z_{-j}}(Z_{-j}^{(t')}, z_{-j}) \tag{55}$$

where $k_{\hat{Y}}, k_{Z_j}, k_{Z_{-j}}$ are the kernels associated with $\mathcal{R}_{\hat{Y}}, \mathcal{R}_{Z_j}$, and $\mathcal{R}_{Z_{-j}}$, respectively.

Following Lemma 2, we conclude

$$\kappa_t^{\text{C-SKIT}} := \tanh(\rho_t^{\text{C-SKIT}}(\hat{Y}, Z_j, Z_{-j}) - \rho_t^{\text{C-SKIT}}(\hat{Y}, \widetilde{Z}_j, Z_{-j})). \tag{56}$$

## B.3   Computing $\rho_t^{\text{X-SKIT}}$ and $\kappa_t^{\text{X-SKIT}}$

Recall that, for a particular sample $z$, a concept $j \in [m]$, and a subset $S \subseteq [m] \setminus \{j\}$ that does not contain $j$, $\rho_t^{\text{X-SKIT}}$ is the estimate—at time $t$—of the witness function of $\text{MMD}(\hat{Y}_{S\cup\{j\}}, \hat{Y}_S)$ with $\hat{Y}_C = g(\widetilde{H}_C)$, $\widetilde{H}_C \sim P_{H|Z_C=z_C}$, i.e.

$$\rho_t^{\text{X-SKIT}} = \hat{\mu}_{\hat{Y}_{S\cup\{j\}}}^{(t-1)} - \hat{\mu}_{\hat{Y}_S}^{(t-1)}, \tag{57}$$

where

$$\hat{\mu}_{\hat{Y}_{S \cup \{j\}}}^{(t-1)} = \frac{1}{t-1} \sum_{t'=0}^{t-1} \varphi_{\hat{Y}}(\hat{Y}_{S \cup \{j\}}^{(t')}), \qquad \hat{\mu}_{\hat{Y}_S}^{(t-1)} = \frac{1}{t-1} \sum_{t'=0}^{t-1} \varphi_{\hat{Y}}(\hat{Y}_S^{(t')}), \qquad (58)$$

and $\varphi_{\hat{Y}}$ is the canonical feature map of $\mathcal{R}_{\hat{Y}}$. Then, for a prediction $\hat{y}$, the value of $\rho_t^{\text{X-SKIT}}$ is

$$\rho_t^{\text{X-SKIT}}(\hat{y}) = (\hat{\mu}_{\hat{Y}_{S \cup \{j\}}}^{(t-1)} - \hat{\mu}_{\hat{Y}_S}^{(t-1)})(\hat{y}) \qquad (59)$$

$$= \hat{\mu}_{\hat{Y}_{S \cup \{j\}}}^{(t-1)}(\hat{y}) - \hat{\mu}_{\hat{Y}_S}^{(t-1)}(\hat{y}) \qquad (60)$$

$$= \frac{1}{t-1} \left[ \sum_{t'=0}^{t-1} k_{\hat{Y}}(\hat{Y}_{S \cup \{j\}}^{(t')}, \hat{y}) - \sum_{t'=0}^{t-1} k_{\hat{Y}}(\hat{Y}_S^{(t')}, \hat{y}) \right], \qquad (61)$$

where $k_{\hat{Y}}$ is the kernel associated with $\mathcal{R}_{\hat{Y}}$.

To conclude, applying Lemma 2 implies

$$\kappa_t^{\text{X-SKIT}} := \tanh(\rho_t^{\text{X-SKIT}}(\hat{Y}_{S \cup \{j\}}) - \rho_t^{\text{X-SKIT}}(\hat{Y}_S)). \qquad (62)$$

## C   Proofs

In this appendix, we include the proofs of the results presented in this paper.

### C.1   Proof of Lemma 1

Recall that $c \in \mathbb{R}^{d \times m}$ is a dictionary of $m$ concepts such that $c_j, j \in [m]$ is the vector representation of the $j^{\text{th}}$ concept. Then, $Z = \langle c, H \rangle$ is the vector where—after appropriate normalization—$Z_j \in [-1, 1]$ represents the amount of concept $j$ in $h$.

We want to show that if $\hat{Y} = \langle w, H \rangle$, $w \in \mathbb{R}^d$, and $d \geq 3$, then

$$\hat{Y} \perp\!\!\!\perp Z_j \;\not\Longleftrightarrow\; \langle w, c_j \rangle = 0. \qquad (63)$$

That is, $w$ being orthogonal to $c_j$ does not provide any information about the statistical dependence between $\hat{Y}$ and $Z_j$, and vice versa.

*Proof.* Herein, for the sake of simplicity, we will drop the $c_j$ notation and consider a single concept $c$. Furthermore, we will assume that all vectors are normalized, i.e. $\|w\| = \|h\| = \|c\| = 1$. Note that the Eq. (63) above can directly be rewritten as

$$\langle w, H \rangle \perp\!\!\!\perp \langle c, H \rangle \;\not\Longleftrightarrow\; \langle w, c \rangle = 0. \qquad (64)$$

( $\not\Longleftarrow$ ) We show there exist random vectors $H$ such that $\langle w, c \rangle = 0$ but $\langle w, H \rangle \not\perp\!\!\!\perp \langle c, H \rangle$.

Let $H \sim \mathcal{U}(\mathbb{S}^d)$, i.e. $H = [H_1, \ldots, H_d]$ is sampled uniformly over the sphere in $d$ dimensions. It is easy to see that $\forall j \in [d]$, $H_j = \langle e_j, H \rangle$, where $e_j$ is the $j^{\text{th}}$ element of the standard basis. Furthermore, it holds that $H_j = \sqrt{1 - \sum_{j' \neq j} H_{j'}^2}$ by definition. We conclude that $\forall (j, j')$, let $w = e_j$ and $c = e_{j'}$, then $\langle w, c \rangle = 0$ but $\langle w, H \rangle \not\perp\!\!\!\perp \langle c, H \rangle$. That is, the fact that $e_j$ and $e_{j'}$ are orthogonal does not mean that their respective projections of $H$ are statistically independent.

( $\not\Longrightarrow$ ) We show how to construct a random vector $H$ such that $\langle w, H \rangle \perp\!\!\!\perp \langle c, H \rangle$ but $\langle w, c \rangle \neq 0$.

Denote $\mathcal{H}_\eta := \{h \in \mathbb{S}^d : \langle c, h \rangle = \eta, \eta \neq 0\}$ the linear subspace of unit vectors in $\mathbb{R}^d$ with the same nonzero inner product with $c$. Each vector $h \in \mathcal{H}_\eta$ can be decomposed into a parallel and an orthogonal component to $c$, i.e. $\forall h \in \mathcal{H}_\eta$, $h = h_c + h_\perp = \eta c + h_\perp$, where the last equality follows by definition of $\mathcal{H}_\eta$.

Consider $\mathcal{H}_\eta$ and $\mathcal{H}_{-\eta}$, it follows that for $w, h^+ \in \mathcal{H}_\eta$ and $h^- \in \mathcal{H}_{-\eta}$

$$\langle w, h^+ \rangle = \langle w, h^- \rangle \iff \langle \eta c + w_\perp, \eta c + h_\perp^+ \rangle = \langle \eta c + w_\perp, -\eta c + h_\perp^- \rangle \tag{65}$$

$$\iff \eta^2 + \langle w_\perp, h_\perp^+ \rangle = -\eta^2 + \langle w_\perp, h_\perp^- \rangle \tag{66}$$

$$\iff \langle w_\perp, h_\perp^+ - h_\perp^- \rangle = -2\eta^2 \tag{67}$$

$$\iff \langle w_\perp, \Delta \rangle = -2\eta^2. \qquad (\Delta := h_\perp^+ - h_\perp^-) \tag{68}$$

Denote $\mathcal{S} = \{(h^+, h^-) : \langle w_\perp, \Delta \rangle = -2\eta^2\}$ the set of pairs of vector that satisfy Eq. (68), and note that for each pair $(h^+, h^-)$ there exists a value $\beta$ such that $\langle w, h^+ \rangle = \langle w, h^- \rangle = \beta$. Then, sampling from $\mathcal{S}$ is equivalent to sampling from the set of pairs of vectors in $\mathcal{H}_\eta$ and $\mathcal{H}_{-\eta}$ that attain the same correlation with $w$.

Note that when $d = 2$, $h_\perp^+, h_\perp^- \in \{\pm w_\perp\}$ by construction, hence $\Delta \in \{0, \pm 2w_\perp\}$ and $\langle w_\perp, \Delta \rangle \in \{0, \pm 2(1 - \eta^2)\}$. Then, $\mathcal{S} = \emptyset$ because there are no pairs of vectors such that $\langle w_\perp, \Delta \rangle = -2\eta^2$. For $d \geq 3$, $\mathcal{S}$ is nonempty as long as $\eta \leq \sqrt{1/2}$.

Then, we can construct $H$ as follows:

- Sample the component parallel to $c$, i.e. $H_c \sim \mathcal{U}(\pm \eta c)$,
- Sample the component orthogonal to $c$, i.e. $(H_\perp^+, H_\perp^-) \sim \mathcal{U}(\mathcal{S})$,

and note that by doing so, we have sampled $\langle c, H \rangle$ and $\langle w, H \rangle$ independently. Finally

$$H = \begin{cases} H_c + H_\perp^+ & \text{if } H_c = \eta c \\ H_c + H_\perp^- & \text{if } H_c = -\eta c \end{cases} \tag{69}$$

has $\langle w, H \rangle \perp\!\!\!\perp \langle c, H \rangle$ by construction, but, since $w \in \mathcal{H}_\eta$, $\langle w, c \rangle = \eta \neq 0$.

This concludes the proof of the lemma. $\qquad \square$

## C.2 Proof of Proposition 1

Recall that Proposition 1 states that the payoff function

$$\kappa_t^{\text{C-SKIT}} = \tanh(\rho_t^{\text{C-SKIT}}(\hat{Y}, Z_j, Z_{-j}) - \rho_t^{\text{C-SKIT}}(\hat{Y}, \widetilde{Z}_j, Z_{-j})) \tag{70}$$

provides a fair game for the global conditional semantic importance null hypothesis $H_{0,j}^{\text{GC}}$ in Definition 2. That is, the wealth process provides Type I error control.

*Proof.* Note that $H_{0,j}^{\text{GC}}$ can be directly rewritten as

$$H_{0,j}^{\text{GC}} : \ P_{\hat{Y} Z_j Z_{-j}} = P_{\hat{Y} \widetilde{Z}_j Z_{-j}} \tag{71}$$

where $\widetilde{Z}_j \sim P_{Z_j | Z_{-j}}$. Then, under the null, the triplets $(\hat{Y}, Z_j, Z_{-j})$ and $(\hat{Y}, \widetilde{Z}_j, Z_{-j})$ are exchangeable by assumption, and the result follows from Lemma 2. $\qquad \square$

## C.3 Proof of Proposition 2

Recall that Proposition 2 claims that a wealth process with

$$\kappa_t^{\text{X-SKIT}} = \tanh(\rho_t^{\text{X-SKIT}}(\hat{Y}_{S \cup \{j\}}) - \rho_t^{\text{X-SKIT}}(\hat{Y}_S)) \tag{72}$$

can be used to reject the local conditional semantic importance $H_{0,j,S}^{\text{LC}}$ in Definition 3 with Type I error control, i.e. the game is fair.

*Proof.* It is easy to see that $H_{0,j,S}^{\text{LC}}$ is already written as a two-sample test. Then, under this null, $\hat{Y}_{S \cup \{j\}}$ and $\hat{Y}_S$ are exchangeable by assumption, and Lemma 2 implies the statement of the proposition. $\qquad \square$

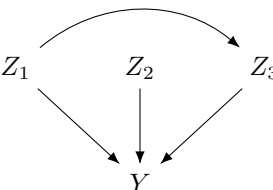

Figure D.1: Pictorial representation of the data-generating process for the synthetic dataset.

## D   Synthetic Experiments

In this section, we showcase the main properties of our tests on two synthetic datasets.

### D.1   Gaussian Data

We start by illustrating all sequential tests presented in this work are valid, and that they adapt to the hardness of the problem, i.e. the weaker the dependence structure, the longer their rejection times. We devise a synthetic dataset with a nonlinear response such that all distributions are known and we can sample from the exact conditional distribution.

The data-generating process we consider is defined as

$$Z_1 \sim \mathcal{N}(\mu_1, \sigma_1^2) \qquad\qquad \mu_1 = 1, \sigma_1 = 1 \tag{73}$$
$$Z_2 \sim \mathcal{N}(\mu_2, \sigma_2^2) \qquad\qquad \mu_2 = -1, \sigma_2 = 1 \tag{74}$$
$$Z_3 \mid Z_1 \sim \mathcal{N}(Z_1, \sigma_3^2) \qquad\qquad \sigma_3 = 1 \tag{75}$$

and

$$Y \mid Z \sim S(\beta_1 Z_1 + \beta_2 Z_2 Z_3 + \beta_3 Z_3) + \epsilon, \tag{76}$$

where $S$ is the sigmoid function, $\epsilon \sim \mathcal{N}(0, \sigma_0^2)$, $\sigma_0 = 0.01$ is independent Gaussian noise, and $\beta_i$, $i = 1, 2, 3$ are coefficients that will allow us to test different conditions. Then, it follows that

$$g(z) = \mathbb{E}[Y \mid Z = z] = S(\beta_1 z_1 + \beta_2 z_2 z_3 + \beta_3 z_3) \tag{77}$$

and

$$Z_1 \mid Z_3 \sim \mathcal{N}\left( \frac{\sigma_1^2}{\sigma_1^2 + \sigma_3^2} Z_3 + \frac{\sigma_3^2}{\sigma_1^2 + \sigma_3^2} \mu_1, \left( \frac{1}{\sigma_1^2} + \frac{1}{\sigma_3^2} \right)^{-1} \right). \tag{78}$$

Fig. D.1 depicts the data-generating process. We remark that, for this experiment, we assume there exists a known parametric relation between the response $Y$ and the *concepts* $Z$. This is only to verify our tests retrieve the ground-truth structure, and our contributions do not rely on this assumption. With this data-generating process, it holds that:

1.  If $\beta_2 = 0$ then $Y \perp\!\!\!\perp Z_2$,

2.  If $\beta_1 = 0$ then $Y \perp\!\!\!\perp Z_1 \mid Z_{-1}$, and

3.  For an observation $Z = z$, if $z_3 = 0$ then $g(\widetilde{Z}_{\{2,3\}}) \stackrel{d}{=} g(\widetilde{Z}_3)$ with $\widetilde{Z}_C \sim P_{Z \mid Z_C = z_C}$.

We test each condition with SKIT, C-SKIT, and X-SKIT, respectively. We use both a linear and RBF kernel with bandwidth set to the median pairwise distance between all previous observations (commonly referred to as the *median heuristic* [27]). For each test, we estimate the *rejection rate* (i.e., how often a test rejects), and the *expected rejection time* (i.e., how many steps of the test it takes to reject) over 100 draws of $\tau^{\max} = 1000$ samples, and with a significance level $\alpha = 0.05$. We remark that a normalized rejection time of 1 means failing to reject in $\tau^{\max}$ steps.

### D.1.1   Global Importance with SKIT

First, we test that

$$\beta_2 = 0 \implies Y \perp\!\!\!\perp Z_2 \tag{79}$$

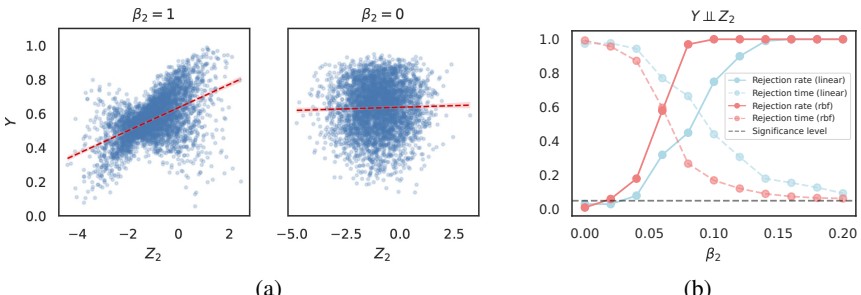

(a)                                                           (b)

Figure D.2: Global importance results for $H_0 : Y \perp\!\!\!\perp Z_2$ with SKIT. (a) Marginal distributions of $Y$ and $Z_2$ for $\beta_2 = 1$ and $0$, respectively. The red dashed line is the linear regression between the two variables, and, as expected, the slope is $\approx 0$ for $\beta_2 = 0$. (b) Mean rejection rate and mean rejection time for SKIT with a linear and RBF kernel, as a function of $\beta_2$.

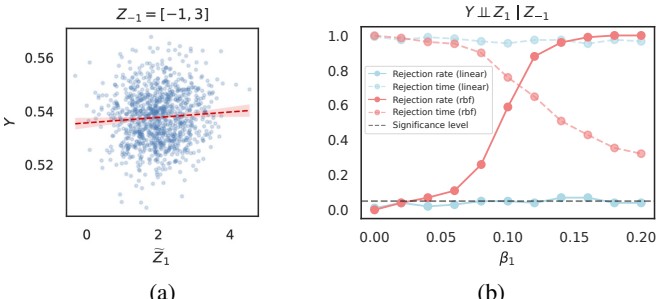

(a)                                           (b)

Figure D.3: Global conditional importance results for $H_0 : Y \perp\!\!\!\perp Z_1 \mid Z_{-1}$ with C-SKIT. (a) $\widetilde{Z}_1 \sim P_{Z_1|Z_{-1}}$ is independent of $Y$ for $Z_{-1} = [-1, 3]$. As expected, the slope of the linear regression between $Y$ and $\widetilde{Z}_1$ is $\approx 0$. (b) Mean rejection rate and mean rejection time for C-SKIT with a linear and RBF kernel, as a function of $\beta_1$.

with the symmetry-based SKIT in Algorithm B.1. Fig. D.2a shows samples from the joint distribution $P_{YZ_2}$ for $\beta_2 = 1$ and $\beta_2 = 0$. As expected, when $\beta_2 = 0$, the slope of the linear regression (red dashed line) is $\approx 0$ because $Y$ and $Z_2$ are independent. Fig. D.2b reports average rejection rate and average rejection time as a function of $\beta_2$. As $\beta_2$ increases, the strength of the dependency between $Y$ and $Z_2$ increases, and the rejection time decreases—this adaptive behavior is characteristic of sequential tests.

We can verify that the rejection rate is below the significance level $\alpha = 0.05$ when $\beta_2 = 0$, and that the SKIT procedure provides Type I error control. Finally, we note that both kernels perform similarly for this test, with the linear kernel generally rejecting less than the RBF one, and with longer rejection times.

### D.1.2 Global Conditional Importance with C-SKIT

Then, we test that

$$\beta_1 = 0 \implies Y \perp\!\!\!\perp Z_1 \mid Z_{-1} \tag{80}$$

with C-SKIT (Algorithm 1). We remark that we can sample from the exact conditional distribution $P_{Z_1|Z_{-1}} = P_{Z_1|Z_{\{2,3\}}}$ because $Z_2$ is independent of $Z_1$ by construction, and the conditional $P_{Z_1|Z_3}$ can be computed analytically as shown in Eq. (78). We verify the conditional distribution behaves as expected in Fig. D.3a. By construction, $\widetilde{Z}_1$ is sampled without looking at $Y$, hence it is independent, and the slope of the linear regression (red dashed line) is $\approx 0$. Fig. D.3b shows mean rejection rate and time as a function of $\beta_1$. First and foremost, we can see that in this case the linear kernel always fails to reject—independently of the value of $\beta_1$. This behavior highlights an important aspect of all kernel-based tests, that is the kernel needs to be *characteristic* in order for the mean embedding to be an injective function [25, 68]. If this condition is not satisfied, different probability distributions could share the same mean embedding in the RKHS, and it may not be possible to disambiguate them

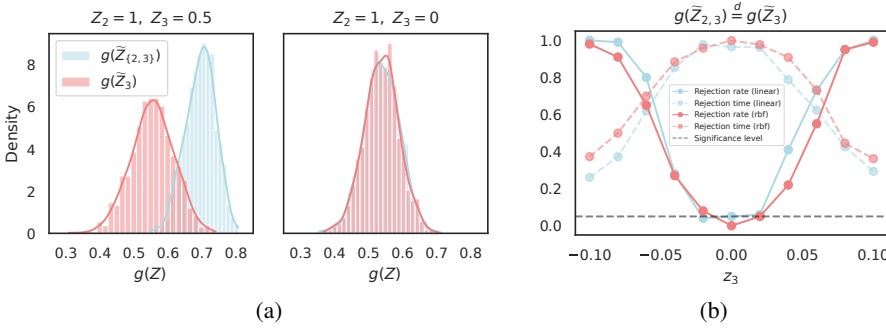

Figure D.4: Local conditional importance results for $H_0 : g(\widetilde{Z}_{\{2,3\}}) \overset{d}{=} g(\widetilde{Z}_3)$ with X-SKIT. (a) Shows that, as expected, the test and null distributions overlap when $z_3 = 0$.

at all. Consequently, the test will not be consistent, and increasing $\tau^{\max}$ will not increase power. For the RBF kernel—which satisfies the characteristic property for probability distributions on $\mathbb{R}^d$—the test is valid (i.e., it provides Type I error control for $\beta_1 = 0$), and it is adaptive to the strength of the conditional dependence structure—as $\beta_1$ increases, the rejection time decreases.

### D.1.3   Local Conditional Importance with X-SKIT

Finally, we test that for a fixed $z$,

$$z_3 = 0 \implies g(\widetilde{Z}_{\{2,3\}}) \overset{d}{=} g(\widetilde{Z}_3), \ \widetilde{Z}_C \sim P_{Z|Z_C = z_C}, \ \forall C \subseteq [m]. \tag{81}$$

with X-SKIT (Algorithm 2). That is—because of the multiplicative term $z_2 z_3$ in $g$—the observed value of $z_2$ does not change the distribution of the response of the model when $z_3 = 0$. Fig. D.4a shows the test (i.e., $g(\widetilde{Z}_{\{2,3\}})$) and null (i.e., $g(\widetilde{Z}_3)$) distributions for different values of $z_3$ when $z_2 = 1$. As expected, we can see that when $z_3 = 0$, the two distributions overlap, whereas when $z_3 = 0.5$, the test distribution is slightly shifted to the right. Fig. D.4b shows results of X-SKIT with both a linear and RBF kernel as a function of $z_3$. We use both positive and negative values of $z_3$ to show that X-SKIT has a two-sided alternative, i.e. it rejects both when the test distribution is to the right and to the left of the null. We can see that both the linear and RBF kernel provide Type I error control when $z_3 = 0$, and that their rejection times adapt to the hardness of the problem.

Now that we have illustrated all tests in arguably the simplest setting, we move onto a synthetic dataset where the response is learned by means of a neural network.

### D.2   Counting MNIST Digits

In this section, we test the semantic importance structure of a neural network trained to count numbers in synthetic images assembled by placing digits from the MNIST dataset [38] in a $4 \times 4$ grid. Fig. D.5 depicts the data-generating process, which satisfies:

- Blue zeros, orange threes, blue twos, and purple sevens are sampled independently with

$$Z_{\text{blue zeros}} \sim \mathcal{U}(\{0, 1, 2\}) + \epsilon \qquad\qquad Z_{\text{orange threes}} \sim \mathcal{U}(\{0, 1, 2\}) + \epsilon \tag{82}$$
$$Z_{\text{blue twos}} \sim \mathcal{U}(\{1, 2\}) + \epsilon \qquad\qquad Z_{\text{purple sevens}} \sim \mathcal{U}(\{1, 2\}) + \epsilon \tag{83}$$

- Green fives are sampled conditionally on blue zeros with

$$Z_{\text{green fives}} \mid Z_{\text{blue zeros}} \sim \text{Cat}\left(\{1, 2, 3\}, \begin{cases} [3/4, 1/8, 1/8] & \text{if } N_{\text{blue zeros}} = 0 \\ [1/8, 3/4, 1/8] & \text{if } N_{\text{blue zeros}} = 1 \\ [1/8, 1/8, 3/4] & \text{if } N_{\text{blue zeros}} = 2 \end{cases}\right) + \epsilon. \tag{84}$$

That is, the number of blue zeros changes the probability distribution of green fives over 1,2,3.

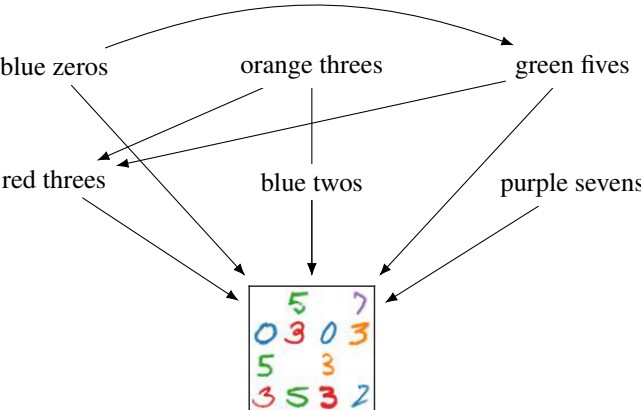

Figure D.5: Pictorial representation of the data-generating process for the counting dataset.

- Red threes are sampled conditionally on both orange threes and green fives with

$$Z_{\text{red threes}} \mid Z_{\text{orange threes}}, Z_{\text{green fives}} \sim 2 + \text{Bernoulli}(p) + \epsilon, \tag{85}$$

$$p = \begin{cases} \alpha & \text{if } N_{\text{orange threes}} N_{\text{green fives}} \geq 3 \\ 1 - \alpha & \text{otherwise} \end{cases}, \; \alpha = 0.9. \tag{86}$$

That is, the product of the number of orange threes and green fives changes the distribution of red threes over 1,2.

Finally, we remark that $N$ denotes the nearest integer to $Z$, and $\epsilon$ is independent uniform noise (i.e., $\epsilon \sim \mathcal{U}(-0.5, 0.5)$) to make the distribution of the concepts continuous. To summarize, in order to generate images, we first sample the concepts $Z$ according to the distribution above, round their values to their respective nearest integers $N$, and finally randomly place digits from the MNIST dataset in a $4 \times 4$ grid according to their number. Note that this data-generating process adds color to the original black and white MNIST digits, and that color matters for the counting task since orange threes and red threes have different distributions.

We remark that, with the data generating process above, we can sample from the true conditional distribution of the digits, and, consequently, of images. We omit details on the conditional distribution for the sake of presentation.[4] We stress that this setting differs slightly from the general one presented in this paper, where we consider both an encoder $f$ and a classifier $g$ such that $\hat{y} = g(f(x))$, and we sample from the conditional distribution of the dense embeddings $H$ given any subset of concepts (i.e., $P_{H|Z_C}$). The scope of this experiment is to showcase the effectiveness of our tests when the response is parametrized by a complex, nonlinear, learned predictor, hence we train a neural network such that $\hat{y} = f(x)$ and directly sample from the conditional distribution of images given any subset of digits (i.e., $P_{X|Z_C}$).

We sample a training dataset of 50,000 images and train a ResNet18 [30] to predict the number of all digits for 6 epochs with batch size of 64 and Adam optimizer [35] with learning rate equal to $10^{-4}$, weight decay of $10^{-5}$, and a scheduler that halves the learning rate every other epoch (recall that the model needs to learn to disambiguate red and orange threes, so color matters). To evaluate the model, we round predictions to the nearest integer and compute accuracy on a held-out set of 10,000 images from the same distribution (we use the original train and test splits of the MNIST dataset to guarantee no digits showed during training are included in test images), and the model achieves an accuracy greater than $99\%$.

Herein, we study the semantic importance structure of the *predicted* number of red threes with respect to the *predicted* number of other digits. Note that the ground-truth distribution satisfies the following conditions:

1. Red threes are independent of blue twos and purple sevens, i.e.

$$Z_{\text{red threes}} \perp\!\!\!\perp Z_{\text{blue twos}} \qquad \text{and} \qquad Z_{\text{red threes}} \perp\!\!\!\perp Z_{\text{purple sevens}}. \tag{87}$$

---

[4] All code necessary to reproduce experiments is available on GitHub.

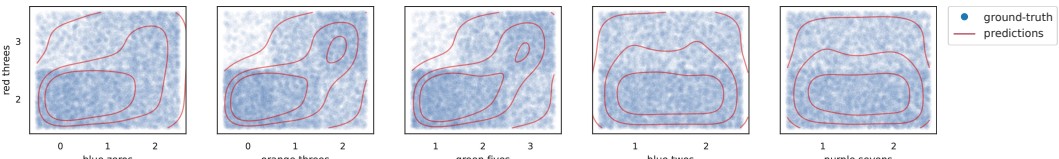

Figure D.6: Distribution of ground-truth data and density estimation of the predictions of the trained model for the validation data in the counting digits experiments.

2. Red threes are independent of blue zeros conditionally on green fives, i.e.

$$Z_{\text{red threes}} \perp\!\!\!\perp Z_{\text{blue zeros}} \mid Z_{\text{green fives}}. \tag{88}$$

3. If—in a specific image—there are no orange threes, then red threes are independent of green fives, i.e.

$$Z_{\text{red threes}} | (N_{\text{green fives}} = n_{\text{green fives}}, N_{\text{orange threes}} = 0) \stackrel{d}{=} Z_{\text{red threes}} | N_{\text{orange threes}} = 0. \tag{89}$$

### D.2.1 Global Importance with SKIT

We start by testing whether the predictions of the model satisfy the ground-truth condition

$$Z_{\text{red threes}} \perp\!\!\!\perp Z_{\text{blue twos}} \qquad \text{and} \qquad Z_{\text{red threes}} \perp\!\!\!\perp Z_{\text{purple sevens}} \tag{90}$$

with SKIT (Algorithm B.1). We remark that at inference, we round predictions to the nearest integer and add independent uniform noise $\epsilon \sim \mathcal{U}(-0.5, 0.5)$ to make the distribution of the response of the model continuous. Fig. D.6 shows the ground-truth distribution of red threes as a function of other digits in the held-out set, and the kernel density estimation of the predictions of the model. As expected, we can see that the ground-truth distribution is marginally dependent on blue zeros, orange threes, and green fives, but it is independent of blue twos and purple sevens.

We repeat all tests 100 times with both linear and RBF kernels with bandwidth set to the median of the pairwise distances of previous observations. We perform tests on independent draws of data of size $\tau^{\max} \in \{100, 200, 400, 800, 1600\}$ from the validation set, and study the rank of importance as a function of $\tau^{\max}$, i.e. the amount of data available to test. Fig. D.7 includes mean rejection rate and mean rejection time for each digit, and the rank of importance of digits as a function of $\tau^{\max}$. We can see that—as expected—both linear and RBF kernels successfully control Type I error for "blue twos" and "purple sevens", and this confirms that the distribution of the predictions of the model agrees with the underlying ground-truth data-generating process. Furthermore, we can see that the rank of importance is stable across different values of $\tau^{\max}$, with purple sevens and blue twos consistently ranked last.

### D.2.2 Global Conditional Importance with C-SKIT

Then, we test whether the predictions of the model satisfy the ground-truth conditional independence condition

$$Z_{\text{red threes}} \perp\!\!\!\perp Z_{\text{blue zeros}} \mid Z_{\text{green fives}} \tag{91}$$

with C-SKIT. Analogous to above, we repeat all tests 100 times with linear and RBF kernels, and Fig. D.8 includes results for $\tau^{\max} \in \{100, 200, 400, 800, 1600\}$. Here—similarly to the synthetic experiment presented in Appendix D.1—we can see that the linear kernel almost always fails to reject, i.e. the mean rejection rates for all digits are close to 0. As discussed earlier, this behavior is due to the fact that the linear kernel is not characteristic for the distributions. On the other hand, the RBF is, and, as expected, it is consistent and it provides Type I error control for the null hypothesis that red threes are independent of blue zeros conditionally on all other digits, which is true. Furthermore, we can see that the rank of importance is less stable compared to the one in Fig. D.2, and in particular, $\tau^{\max} = 100$ seems not to be sufficient to retrieve the correct ground-truth structure (i.e., blue twos are ranked before green fives). This highlights how the amount of data available for testing may affect results and findings.

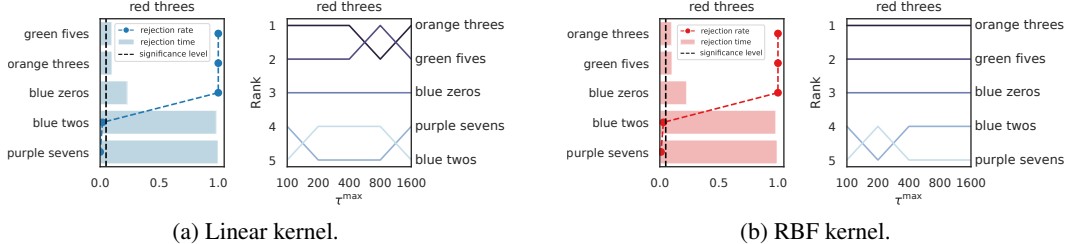

(a) Linear kernel.                                    (b) RBF kernel.

Figure D.7: Global semantic importance results for the predicted number of red threes with linear and RBF kernels. In each subfigure, the leftmost panel shows mean rejection rate and mean rejection time over 100 tests with $\alpha = 0.05$ and $\tau^{\max} = 800$. The rightmost panel shows the rank of importance of digits for the prediction of red threes as a function of $\tau^{\max}$.

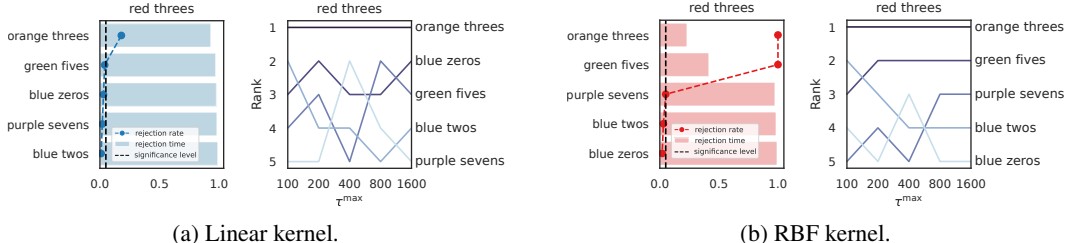

(a) Linear kernel.                                    (b) RBF kernel.

Figure D.8: Global conditional semantic importance results for the predicted number of red threes with linear and RBF kernels. In each subfigure, the leftmost panel shows mean rejection rate and mean rejection time over 100 tests with $\alpha = 0.05$ and $\tau^{\max} = 800$. The rightmost panel shows the rank of importance of digits for the prediction of red threes as a function of $\tau^{\max}$.

### D.2.3 Local Conditional Importance with X-SKIT

Finally, we test whether the predictions of the model satisfy the ground-truth condition that, for a particular image, if there are no orange threes (i.e., $n_{\text{orange threes}} = 0$) then red threes are independent of the observed green fives (i.e., $n_{\text{green fives}}$), i.e.

$$Z_{\text{red threes}}|(N_{\text{green fives}} = n_{\text{green fives}}, N_{\text{orange threes}} = 0) \overset{d}{=} Z_{\text{red threes}}|N_{\text{orange threes}} = 0. \qquad (92)$$

We remark that, in the equation above, conditioning is written in terms of the integer $n_{\text{orange threes}} = 0$ because of its intuitive meaning, and that this is equivalent to conditioning on $z_{\text{orange threes}} \in (-0.5, 0.5)$. Similarly, we could replace $n_{\text{green fives}}$ with $z_{\text{green fives}}$, and, in practice, we run tests conditioning on the observed concepts $z$, and not their integer values $n$.

We use X-SKIT (Algorithm 2) with a linear and RBF kernel with bandwidth set to the median of the pairwise distances of previous observations. We repeat all tests 100 times on individual images with 0, 1, and 2 orange threes, significance level $\alpha = 0.05$, and $\tau^{\max} = 400$. Fig. D.9 shows results grouped by number of orange threes. As expected, we see that when $n_{\text{orange threes}}$ is grater than 0, the number of green fives is important for the predictions of the model (i.e., rejection rate is close to 1, with short rejection rate), whereas when there are no orange threes in the image, both the linear and RBF kernel control Type I error. We qualitatively compared our findings with pixel-level explanations with Grad-CAM [57], and we can see that they only highlight red threes because that is the digit we are explaining the prediction of. That is, pixel-level explanations cannot convey the full spectrum of semantic importance for the predictions of a model—which can be misleading to users. For example, in this case, a user may not understand when the predictions of a model depend on the number of green fives, because they are never highlighted by pixel-level saliency maps. In real-world scenarios, digits may be replaced by sensitive attributes that cannot be inferred by the raw value of pixels. For example, a saliency map highlighting a face does not convey which attributes were used by the model, such as skin color, biological sex, or gender. It is immediate to see how being able to investigate the dependencies of the predictions of a model with respects to these attributes (which our definitions provide) is paramount for their safe deployment.

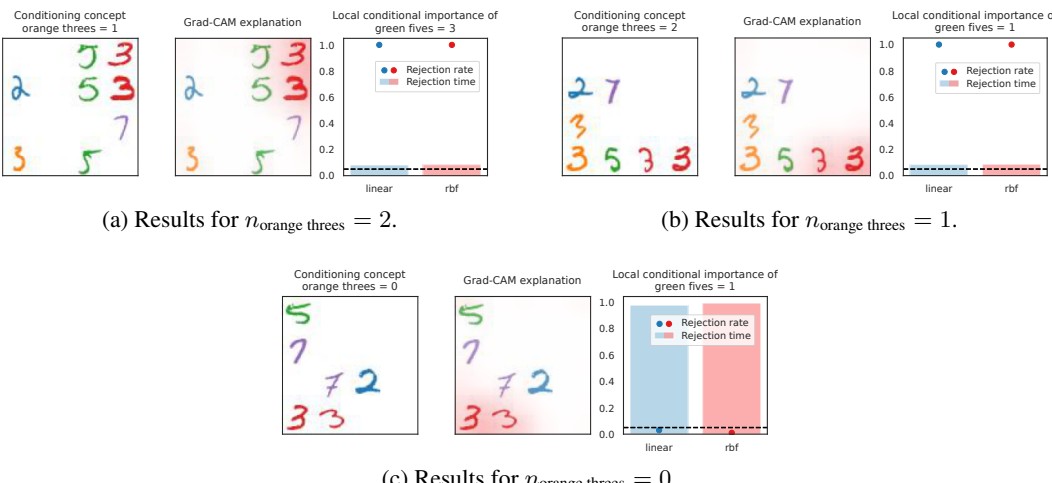

(a) Results for $n_{\text{orange threes}} = 2$.          (b) Results for $n_{\text{orange threes}} = 1$.

(c) Results for $n_{\text{orange threes}} = 0$.

Figure D.9: Local conditional importance of $N_{\text{green fives}}$ conditionally on $N_{\text{orange threes}}$. Each row contains the input image, the Grad-CAM explanation for the prediction of the model, and X-SKIT results for 100 repetitions of the test with $\tau^{\max} = 400$, with a linear and RBF kernel. Note that X-SKIT finds the observed number of green fives important whenever the number of orange threes is greater than zero, whereas Grad-CAM does not.

With these results on synthetic datasets, we now showcase the flexibility of our proposed tests on zero-shot image classification with several and diverse vision-language (VL) models.

# E    Experimental Details

In this appendix, we include further details about the real-world experiments that were omitted from the main text for the sake of presentation. All experiments were run on a private server with one 24 GB NVIDIA RTX A5000 GPU and 96 CPU cores with 500 GB of RAM memory.

**List of VL models used in the experiments.** We use 8 different VL models, both CLIP- and non-CLIP-based: CLIP:RN50,ViT-B/32,ViT-L/14 [52], OpenClip:ViT-B-32,ViT-L-14 [31], FLAVA [65], ALIGN [32], and BLIP [39].

**Evaluating rank agreement.** We use a weighted version of Kendall's tau [33] introduced by Vigna [74] which assigns higher penalties to swaps between elements with higher ranks. This choice reflects the fact that concepts with higher importance should be more stable across different models. We briefly remark that this notion of rank agreement is bounded in $[-1, 1]$ ($-1$ indicates reverse order, and 1 perfect alignment) but not symmetric.

**Evaluating importance agreement.** We threshold rejection rates at level $\alpha$ to classify concepts into important and not important ones. Then, importance agreement is the accuracy between pairs of binarized vectors.

## E.1    Estimating Conditional Distributions from Data

Here, we introduce nonparametric methods to estimate the conditional distributions necessary to run our C-SKIT and X-SKIT tests. Throughout this section, we will assume access to a training set $\{(h^{(i)}, z^{(i)})\}_{i=1}^{n}$ of $n$ tuples of dense embeddings $h \in \mathbb{R}^d$ with their semantics $z \in [-1, 1]^m$.

### E.1.1    Estimating $P_{Z_j | Z_{-j}}$ for C-SKIT

Here, we describe how to sample from the conditional distribution of a concept $Z_j$ given the rest, $Z_{-j}$, i.e. $\widetilde{Z}_j \sim P_{Z_j | Z_{-j}}$, which is necessary to run our C-SKIT test. In particular, for a concept

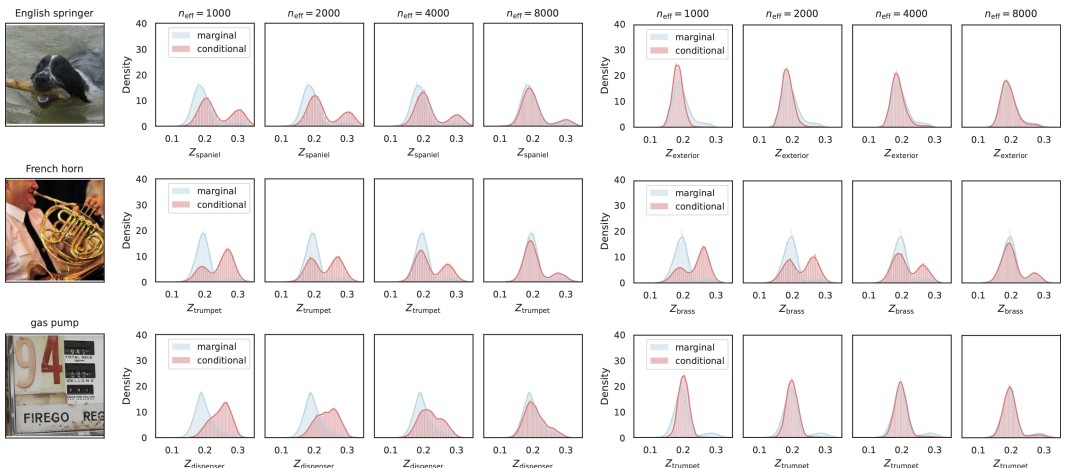

Figure E.1: Example marginal and estimated conditional distributions $p(z_j)$ and $\tilde{p}(z_j|z_{-j})$ for two class-specific concepts on three images from the Imagenette dataset. Distributions are shown as a function of the effective number of points in the weighted KDE (i.e., $n_{\text{eff}}$).

$j \in [m]$, and an observation $z \in [-1, 1]^m$, we define the unnormalized conditional distribution

$$\tilde{p}(z_j \mid z_{-j}) = \sum_{i=1}^{n} w_i \phi \left( \frac{z_j^{(i)} - z_j}{\nu_{\text{scott}}} \right) \tag{93}$$

by means of weighted kernel density estimation (KDE), where $\phi$ is the standard normal probability density function, $\nu_{\text{scott}}$ is Scott's factor [56], and

$$w_i = \phi \left( \frac{z_{-j}^{(i)} - z_{-j}}{\nu} \right), \; \nu > 0 \tag{94}$$

are the weights. That is, the further $z_{-j}^{(i)}$ is from $z_{-j}$, the lower its weight in the KDE. As for all kernel-based methods, the bandwidth $\nu$ is important for the practical performance of the model. For our experiments, we choose $\nu$ adaptively such that the effective number of points in the KDE (i.e. $n_{\text{eff}} = (\sum_{i=1}^{n} w_i)^2 / \sum_{i=1}^{n} w_i^2$) is the same across concepts. This choice is motivated by the fact that different concepts have different distributions, and we want to guarantee the same number of points are used to estimate their conditional distributions. Furthermore, we note that $n_{\text{eff}}$ controls the strength of the conditioning—the larger $n_{\text{eff}}$, the slower the decay of the weights, and the weaker the conditioning. That is, in the limit, the weights become uniform, the conditional distribution tends to the marginal $\tilde{p}(z_j)$, and all tests presented become of decorrelated semantic importance [72, 73]. With this, sampling $\widetilde{Z}_j \sim P_{Z_j|Z_{-j}=z_{-j}}$ boils down to first sampling $i'$ according to the weights $w = [w_i, \ldots, w_n]$, and then sampling $\widetilde{Z}_j$ from the Gaussian distribution centered at $z_j^{(i')}$.

Fig. E.1 shows the marginal (i.e., $p(z_j)$) and estimated conditional distributions (i.e., $\tilde{p}(z_j|z_{-j})$) of two class-specific concepts as a function of effective number of points $n_{\text{eff}}$ for three images in the Imagenette dataset. We can see how as $n_{\text{eff}}$ increases, the estimated conditional distribution becomes closer to the marginal, and that the conditional distributions of class-specific concepts tend to be skewed to higher values compared to their marginals. This behavior suggests that images from a specific class have higher values of concepts that are related to the class. We use $n_{\text{eff}} = 2000$ for all tests across all real-world experiments.

### E.1.2 Estimating $P_{H|Z_C=z_C}$ for x-SKIT

Here, we describe how to sample from the conditional distribution of dense embeddings $H$ conditionally on any subset of concepts $C \subseteq [m]$ of a particular semantic vector $z \in [-1, 1]^m$, i.e. $\widetilde{H}_C \sim P_{H|Z_C=z_C}$, which is necessary to run our x-SKIT test. We show how to achieve this by coupling the nonparametric sampler introduced above with ideas of nearest neighbors. This choice is

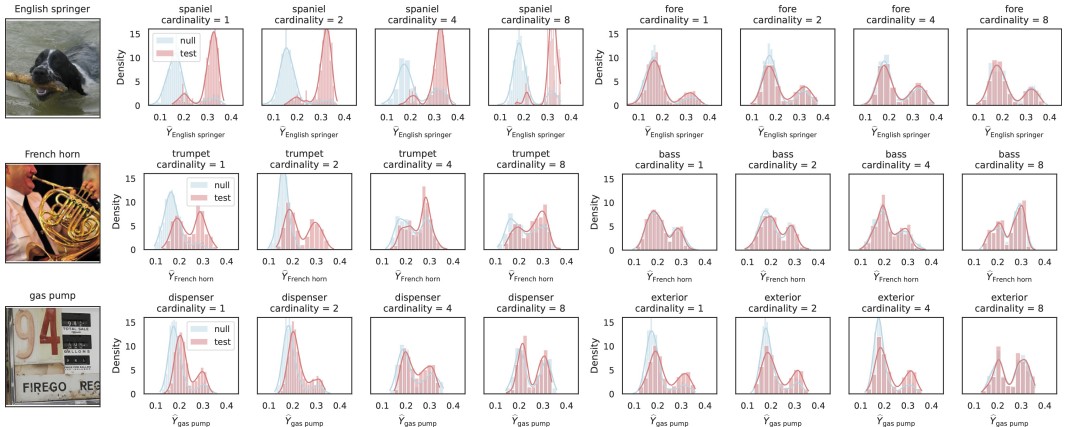

Figure E.2: Example test (i.e., $g(\widetilde{H}_{S\cup\{j\}})$) and null (i.e., $g(\widetilde{H}_S)$) distributions for a class-specific concept and a non-class specific one on three images from the Imagenette dataset as a function of the cardinality of $S$.

motivated by the need to keep samples in-distribution with respects to the downstream classifier $g$. Intuitively, we propose to:

1. Sample $\widetilde{Z} \sim P_{Z|Z_C=z_C}$, and then

2. Retrieve the embedding $H^{(i')}$ such that its concept representation $Z^{(i')}$ is the nearest neighbor of $\widetilde{Z}$.

Step 2 makes sure that samples are coming from *real images*, and it overcomes some of the hurdles of sampling a high-dimensional vector ($H \in \mathbb{R}^d$, $d \sim 10^2$) conditionally on a low-dimensional one ($z_C \in [-1, 1]^{|C|}$, $C \subseteq [m]$, $m \approx 20$). More precisely, recall that $\{(h^{(i)}, z^{(i)})\}_{i=1}^n$ is a set of $n$ pairs of dense embeddings and their semantics, then

$$\widetilde{H}_C = h^{(i')} \quad \text{s.t.} \quad i' = \arg\min_{i\in[n]} \|z^{(i)} - \widetilde{Z}\|, \ \widetilde{Z} \sim P_{Z|Z_C=z_C}, \tag{95}$$

where $P_{Z|Z_C=z_C}$ is approximated with $\tilde{p}(z_{-C} \mid z_C)$, $-C := [m] \setminus C$ as in Eq. (93).

Fig. E.2 shows some example test (i.e., $g(\widetilde{H}_{S\cup\{j\}})$) and null distributions (i.e., $g(\widetilde{H}_S)$) for a class-specific concept and a non-class specific one on the same three images from the Imagenette dataset as in Fig. E.1. We remark that $S$ can be any subset of the remaining concepts, so we show results for random subsets of increasing size. We can see that the test distributions of class-specific concepts are skewed to the right, i.e. including the observed class-specific concept increases the output of the predictor. Furthermore, we see the shift decreases the more concepts are included in $S$, i.e. if $S$ is larger and it contains more information, then the marginal contribution of one additional concept will be smaller. On the other hand, including a non-class-specific concept does not change the distribution of the response of the model, no matter the size of $S$—precisely as our local definition of importance ($H_{0,j,S}^{\text{LC}}$) demands.

## E.2   AwA2 Dataset

Here, we include additional information, tables, and figures for the AwA2 dataset experiment. This dataset comprises 37,322 images (29,841 for training and 7,481 for testing) from 50 animal species along with class-level annotations of 85 attributes (some example figures are included in Fig. E.3). Concept annotations are reported both as frequencies (i.e., how often an attribute appears in images coming from a class) and as binary labels (i.e., 1 means that an attribute is present in a class, and 0 otherwise). Table E.2 shows the zero-shot classification performance of all VL models used in this experiment for the top-10 classes, and, for each class, we test 20 attributes: the 10 most frequent, and a random subset of 10 absent ones (concepts are included in Table E.3). Finally we remark that, for each model, we obtain the dictionary $c$ by encoding concepts with its text encoder.

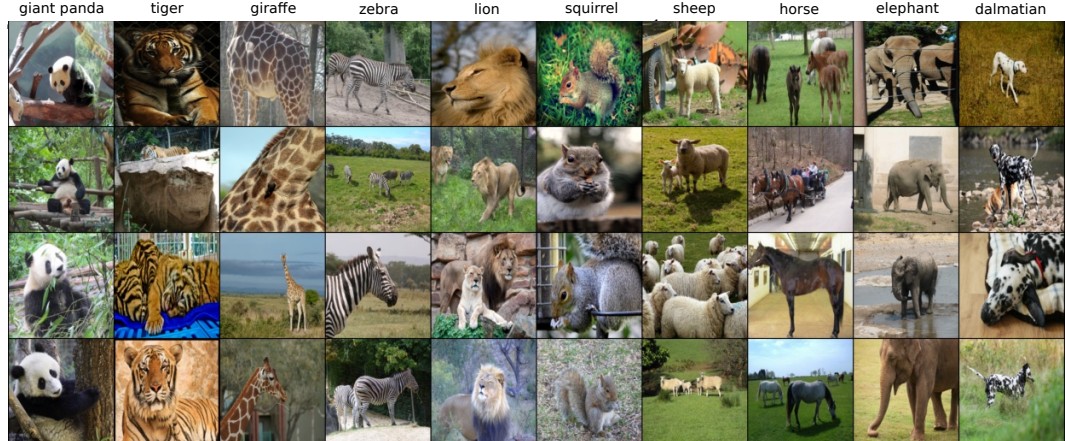

Figure E.3: Example images from the top-10 best classified classes in the AwA2 dataset.

Table E.2: Zero-shot classification accuracy on the AwA2 dataset.

| Model | Accuracy | | | | |
|---|---|---|---|---|---|
| | giant panda | tiger | giraffe | zebra | lion |
| CLIP:RN50 | 100.00% | 100.00% | 100.00% | 100.00% | 99.51% |
| CLIP:ViT-B/32 | 100.00% | 100.00% | 100.00% | 100.00% | 99.51% |
| CLIP:ViT-L/14 | 100.00% | 100.00% | 99.59% | 100.00% | 100.00% |
| OpenClip:ViT-B-32 | 100.00% | 100.00% | 100.00% | 100.00% | 100.00% |
| OpenClip:ViT-L-14 | 100.00% | 100.00% | 100.00% | 100.00% | 100.00% |
| FLAVA | 100.00% | 98.86% | 100.00% | 100.00% | 99.02% |
| ALIGN | 100.00% | 100.00% | 100.00% | 99.57% | 100.00% |
| BLIP | 100.00% | 100.00% | 99.17% | 99.15% | 99.51% |
| average | 100.00% ± 0.00 | 99.86% ± 0.38 | 99.84% ± 0.29 | 99.84% ± 0.30 | 99.69% ± 0.34 |

| Model | Accuracy | | | | |
|---|---|---|---|---|---|
| | squirrel | sheep | horse | elephant | dalmatian |
| CLIP:RN50 | 99.17% | 97.54% | 98.18% | 94.71% | 96.36% |
| CLIP:ViT-B/32 | 99.58% | 98.59% | 99.39% | 99.04% | 98.18% |
| CLIP:ViT-L/14 | 100.00% | 99.65% | 98.78% | 100.00% | 100.00% |
| OpenClip:ViT-B-32 | 99.58% | 99.65% | 98.78% | 100.00% | 97.27% |
| OpenClip:ViT-L-14 | 100.00% | 100.00% | 100.00% | 100.00% | 100.00% |
| FLAVA | 99.17% | 100.00% | 99.39% | 99.04% | 98.18% |
| ALIGN | 99.58% | 99.65% | 99.70% | 100.00% | 99.09% |
| BLIP | 99.17% | 98.94% | 99.39% | 100.00% | 100.00% |
| average | 99.53% ± 0.33 | 99.25% ± 0.80 | 99.20% ± 0.55 | 99.10% ± 1.71 | 98.64% ± 1.29 |

Table E.3: Class-level attributes tested on the AwA2 dataset.

| Class | Attributes (20) | |
|---|---|---|
| | present (10) | absent (10) |
| giant panda | patches, old world, furry, black, big, white, walks, paws, bulbous, vegetation | flies, flippers, hooves, desert, hairless, red, blue, horns, plankton, yellow |
| tiger | stripes, stalker, meat, meat teeth, hunter, strong, fierce, old world, muscle, big | strain teeth, tunnels, hops, plankton, bipedal, tusks, flippers, flies, small, skimmer |
| giraffe | long neck, long legs, big, quadrupedal, spots, vegetation, lean, old world, walks, ground | bipedal, hibernate, cave, mountains, ocean, hunter, water, stripes, gray, fish |
| zebra | stripes, old world, quadrupedal, black, white, ground, group, grazer, walks, hooves | blue, spots, brown, water, coastal, patches, tusks, claws, scavenger, red |
| lion | meat, stalker, strong, hunter, meat teeth, big, fierce, paws, furry, old world | tunnels, blue, horns, skimmer, long neck, water, flippers, tusks, arctic, spots |
| squirrel | tail, furry, forager, small, gray, tree, new world, forest, vegetation, brown | horns, blue, yellow, tusks, meat teeth, flippers, scavenger, desert, plankton, strain teeth |
| sheep | white, quadrupedal, walks, group, vegetation, grazer, ground, fields, furry, new world | arctic, flippers, insects, paws, long neck, red, yellow, swims, plankton, hands |
| horse | hooves, fast, grazer, big, long legs tail, quadrupedal, fields, brown, strong | arctic, tree, bipedal, plankton, fish, stripes, ocean, strain teeth, scavenger, orange |
| elephant | big, old world, gray, tough skin, quadrupedal tusks, hairless, strong, ground, walks | claws, flippers, orange, swims, ocean stripes, tunnels, plankton, coastal, strain teeth |
| dalmatian | big, old world, gray, tough skin, quadrupedal tusks, hairless, strong, ground, walks | claws, flippers, orange, swims, ocean stripes, tunnels, plankton, coastal, strain teeth |

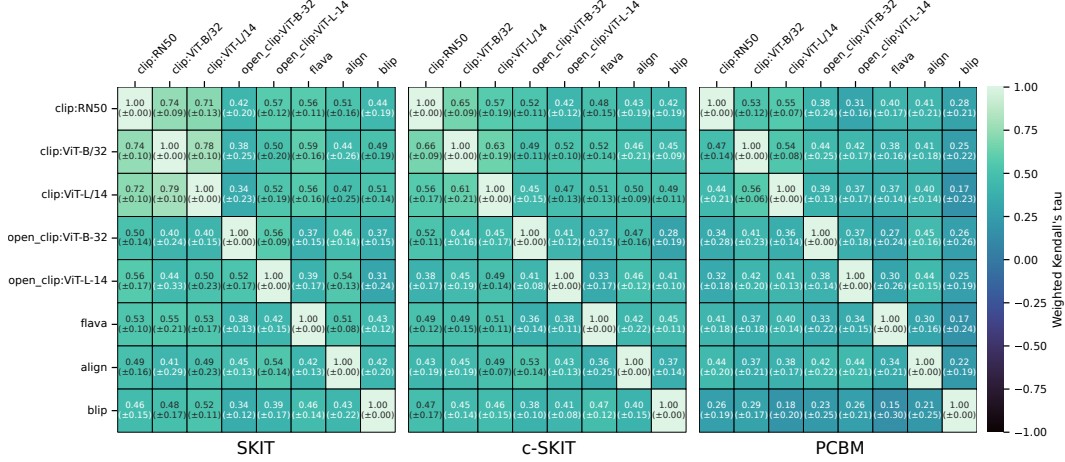

Figure E.4: Rank agreement comparison between SKIT, C-SKIT, and PCBM on the AwA2 dataset. Results are reported as mean and standard deviation over the 10 classes considered in this experiment.

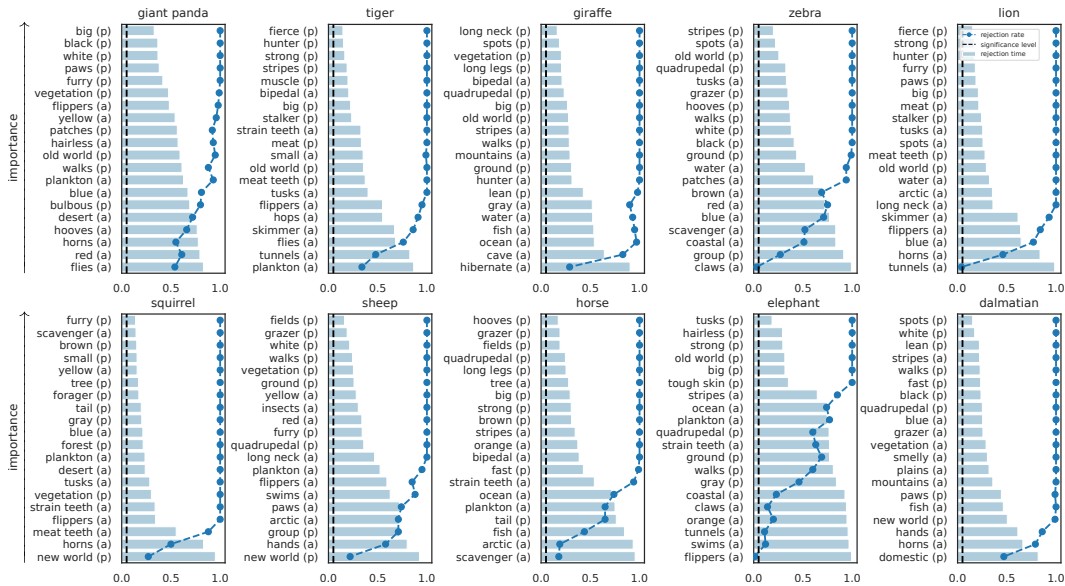

Figure E.5: SKIT importance results with CLIP:ViT-L/14 on the AwA2 dataset.

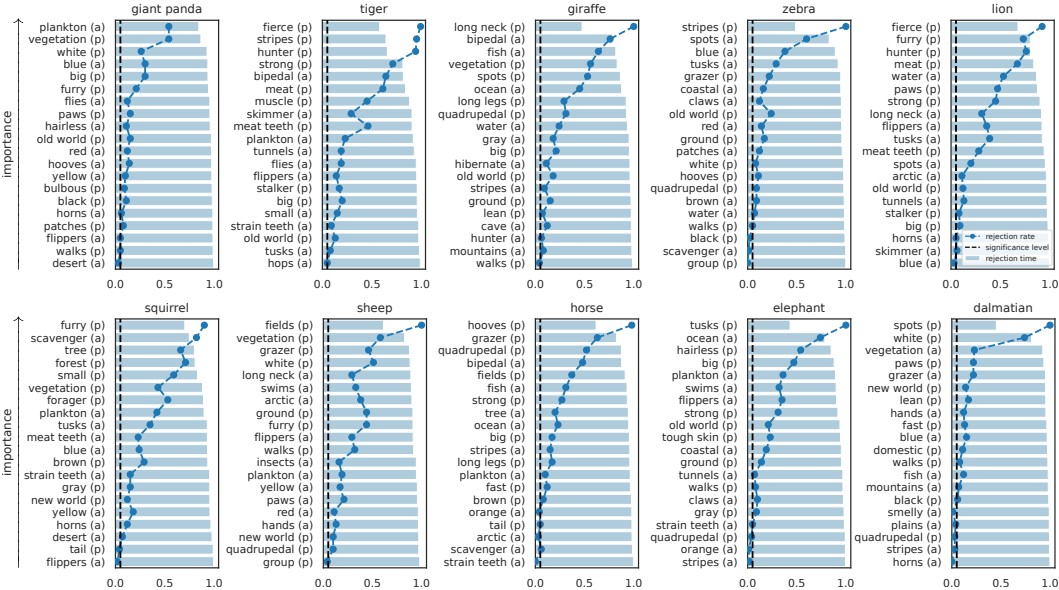

Figure E.6: C-SKIT importance results with CLIP:ViT-L/14 on the AwA2 dataset.

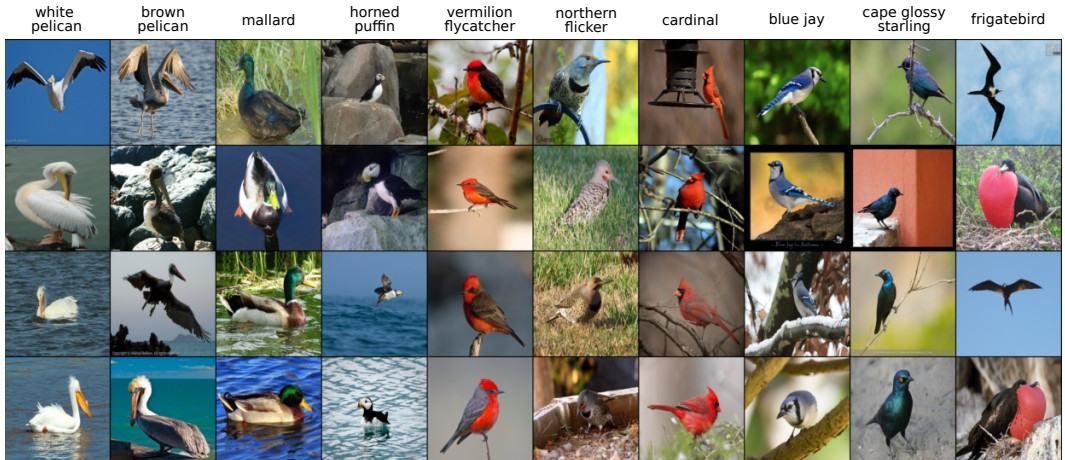

Figure E.7: Example test images from the CUB dataset with their respective classes.

Table E.4: Zero-shot classification accuracy on the CUB dataset.

| Model | Accuracy | | | | |
|---|---|---|---|---|---|
| | white pelican | brown pelican | mallard | horned puffin | vermilion flycatcher |
| CLIP:RN50 | 92.00% | 95.00% | 95.00% | 86.67% | 88.33% |
| CLIP:ViT-B/32 | 98.00% | 96.67% | 90.00% | 93.33% | 93.33% |
| CLIP:ViT-L/14 | 98.00% | 100.00% | 100.00% | 95.00% | 100.00% |
| OpenClip:ViT-B-32 | 100.00% | 98.33% | 98.33% | 88.33% | 96.67% |
| OpenClip:ViT-L-14 | 100.00% | 100.00% | 100.00% | 96.67% | 100.00% |
| FLAVA | 100.00% | 100.00% | 98.33% | 95.00% | 91.67% |
| ALIGN | 96.00% | 96.67% | 95.00% | 98.33% | 78.33% |
| BLIP | 98.00% | 80.00% | 96.67% | 65.00% | 55.00% |
| average | $97.75\% \pm 2.54\%$ | $95.83\% \pm 6.24\%$ | $96.67\% \pm 3.12\%$ | $89.79\% \pm 10.08\%$ | $87.92\% \pm 14.09\%$ |

| Model | Accuracy | | | | |
|---|---|---|---|---|---|
| | northern flicker | cardinal | blue jay | cape glossy starling | frigatebird |
| CLIP:RN50 | 78.33% | 63.16% | 65.00% | 73.33% | 78.33% |
| CLIP:ViT-B/32 | 98.33% | 71.93% | 83.33% | 90.00% | 80.00% |
| CLIP:ViT-L/14 | 98.33% | 100.00% | 85.00% | 100.00% | 96.67% |
| OpenClip:ViT-B-32 | 96.67% | 100.00% | 88.33% | 96.67% | 88.33% |
| OpenClip:ViT-L-14 | 100.00% | 100.00% | 91.67% | 95.00% | 93.33% |
| FLAVA | 76.67% | 100.00% | 91.67% | 88.33% | 90.00% |
| ALIGN | 86.67% | 92.98% | 83.33% | 80.00% | 55.00% |
| BLIP | 61.67% | 61.40% | 90.00% | 73.33% | 75.00% |
| average | $87.08\% \pm 12.96\%$ | $86.18\% \pm 16.42\%$ | $84.79\% \pm 8.14\%$ | $87.08\% \pm 9.75\%$ | $82.08\% \pm 12.47\%$ |

## E.3  CUB Dataset

Here, we include additional information and results for the experiment of the CUB dataset [77], which contains 11,788 images of 200 different bird classes, and each image is annotated with the presence of 312 fine-grained concepts that describe the appearance of the bird (e.g., "has orange bill", "has hook-shaped bill", "is small") with the labelers' confidence. Formally, the dataset is a collection $\{(x^{(i)}, y^{(i)}, z^{(i)}, u^{(i)})\}_{i=1}^{n}$ of images $x$ with class label $y$, binary semantic vector $z^{(i)} \in \{0,1\}^m$, and uncertainty values $u^{(i)} \in \{1,2,3,4\}$: "not visible" (1), "guessing" (2), "probably" (3), and "definitely" (4).

We randomly sample 10 images from the 10 classes with highest average accuracy across models: Table E.4 includes accuracies of all VL models used, and Fig. E.7 shows some example test images

Table E.5: X-SKIT results on the CUB dataset as a function of conditioning set size $s$.

| $s$ | Rank agreement | Importance agreement | Importance $f_1$ score |
|---|---|---|---|
| 1 | $0.81 \pm 0.14$ | $0.96\% \pm 0.06\%$ | $0.93 \pm 0.15$ |
| 2 | $0.82 \pm 0.13$ | $\mathbf{0.97\% \pm 0.06\%}$ | $\mathbf{0.93 \pm 0.14}$ |
| 4 | $\mathbf{0.84 \pm 0.12}$ | $0.95\% \pm 0.08\%$ | $0.88 \pm 0.15$ |

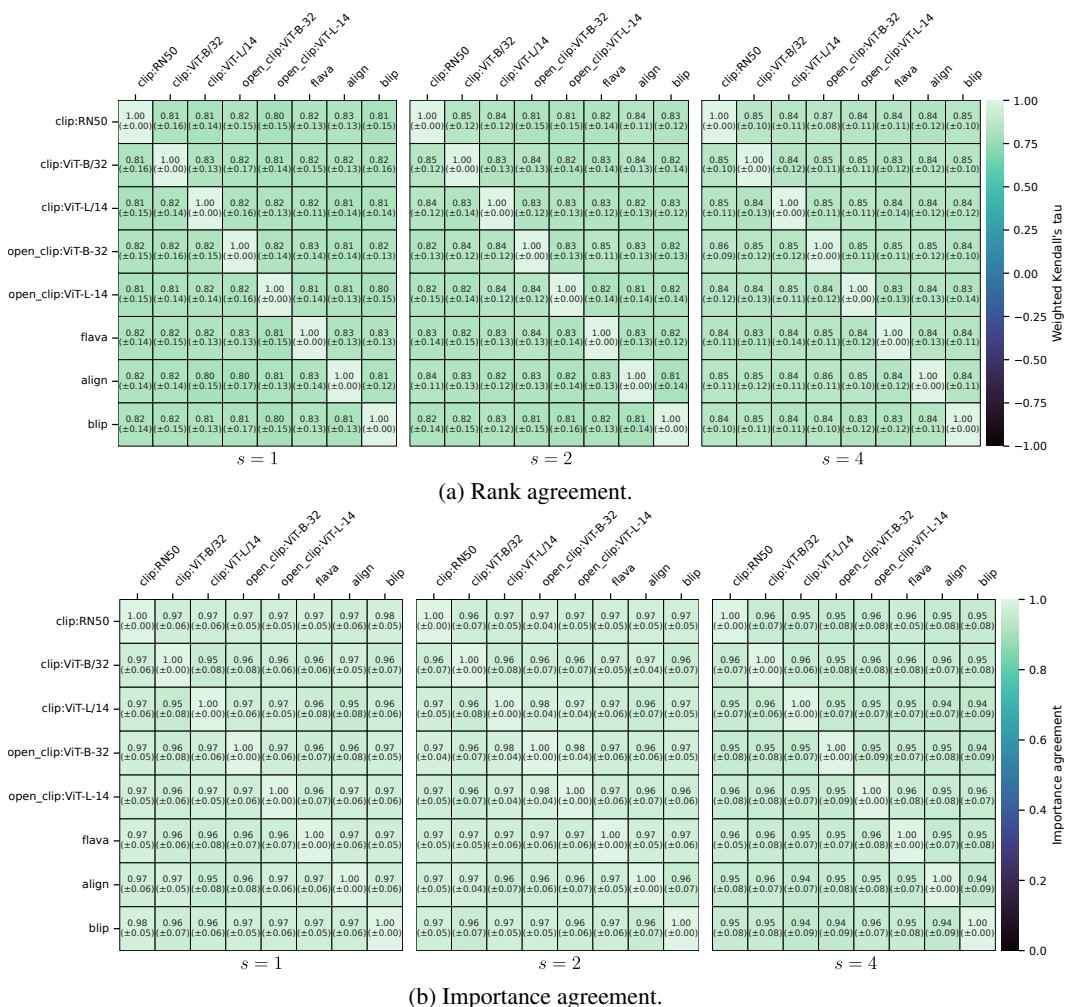

(a) Rank agreement.

(b) Importance agreement.

Figure E.8: X-SKIT agreement results on the CUB dataset as a function of conditioning set size, $s$. Results are reported as means and standard deviations over the random 100 images used in the experiment.

for each class. For each image, we select 14 concepts to test: we first restrict ourselves to annotations with good confidence (i.e., "probably" or "definitely"), and then, for each concept $j$, we estimate its marginal (i.e., $p_j := \mathbb{P}[Z_j = 1]$) and class-conditional (i.e., $p_{j|y} := P[Z_j = 1 \mid Y = y]$) rates over the dataset. Finally, for each test image $x$ with label $y$, we score concepts by the difference between their class-conditional and marginal rates, i.e. $s_j(y) = p_{j|y} - p_j$. Intuitively, a large value of $s_j(y)$ indicates that concept $j$ has a higher occurrence in class $y$ compared to the population, and we say that it is *discriminative* for class $y$. We test the 7 most discriminative concepts that are present in the observed image $x$, and a random subset of 7 concepts that are absent according to the ground-truth annotations. We remark that, since concepts are binary, we do not use the KDE-based methods presented in Appendix E.1, and instead we approximate $P_{H|Z_C = z_C}$ by sampling uniformly from the entries in the dataset that match the conditioning vector $z_C$. Finally, we use RBF kernels with bandwidths set to the median of the pairwise distances of observations, and $\tau^{\max} = 200$.

After running X-SKIT, we classify concepts as important by thresholding their rejection rates at level $\alpha$—which is a statistically-valid way of selecting important concepts. Table E.5 summarizes agreement and detection results as a function of conditioning set size $s$ (i.e., the number of concepts in $S \subseteq [m] \setminus \{j\}$), and Fig. E.8 includes all pairwise agreement values. We note that the $f_1$ score for $s = 4$ (i.e., when conditioning on 4 concepts) is lower compared to $s = 1, 2$. This is expected, as the more concepts one conditions on, the smaller the effect of including one additional concept.

Finally, Fig. E.9 shows ranks of importance for all models on the 4 example images used in the main body of the paper.

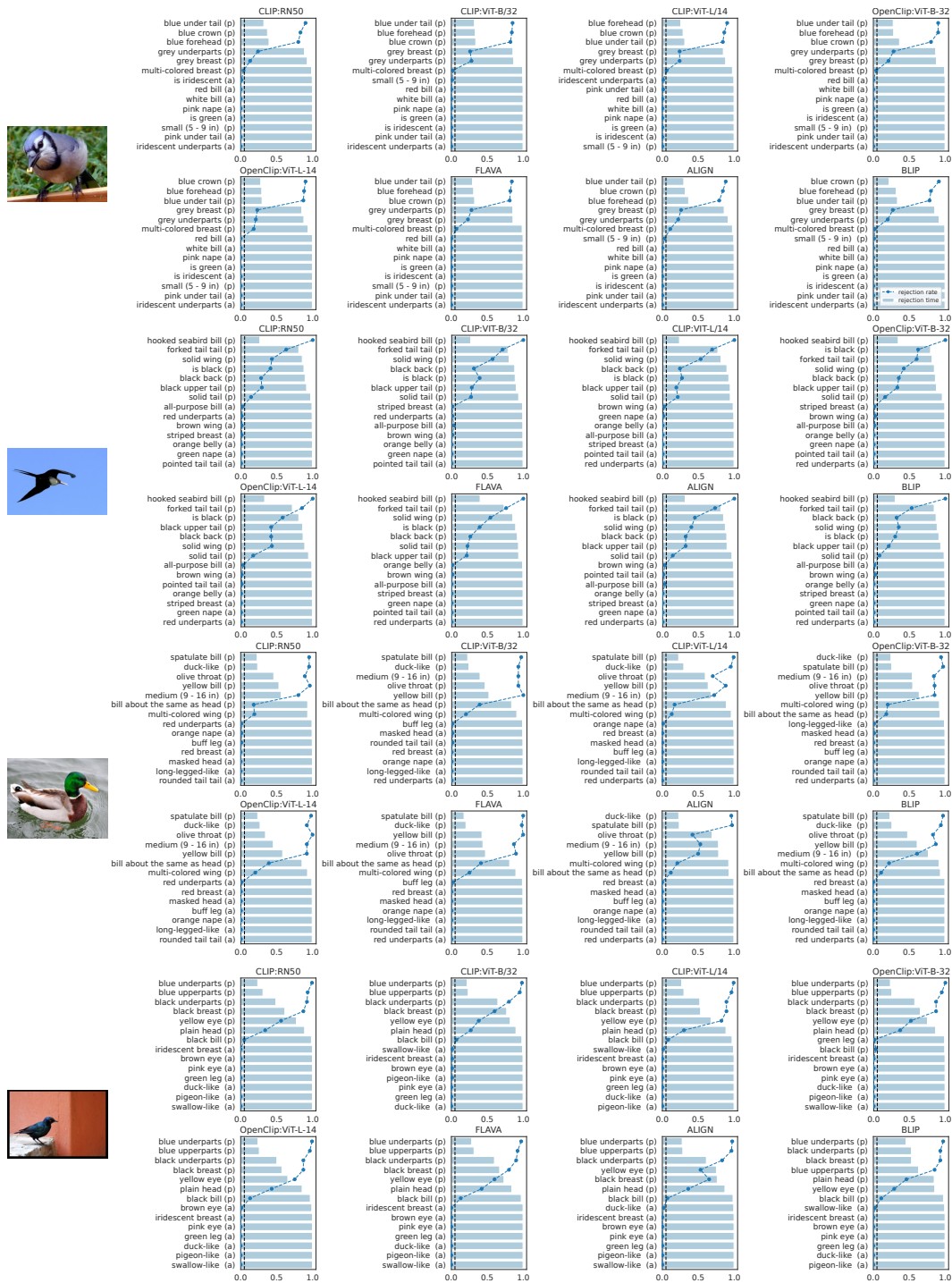

Figure E.9: x-SKIT importance results ($s = 1$) across all models for four example images in the CUB dataset.

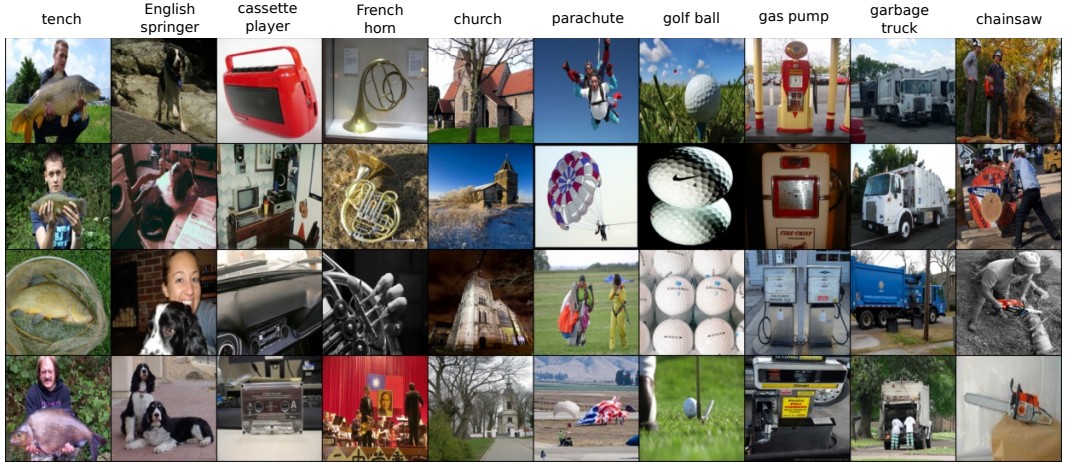

Figure E.10: Example images from the Imagenette dataset with their respective labels.

Table E.6: Zero-shot classification accuracy on the Imagenette dataset.

| Model | Accuracy | | | | |
| --- | --- | --- | --- | --- | --- |
| | tench | English springer | cassette player | French horn | church |
| CLIP:RN50 | 99.48% | 99.49% | 95.80% | 99.49% | 100.00% |
| CLIP:ViT-B/32 | 99.74% | 99.24% | 96.08% | 98.73% | 100.00% |
| CLIP:ViT-L/14 | 99.22% | 99.75% | 99.72% | 99.75% | 100.00% |
| OpenClip:ViT-B-32 | 100.00% | 100.00% | 98.32% | 99.24% | 99.76% |
| OpenClip:ViT-L-14 | 100.00% | 100.00% | 99.72% | 99.75% | 100.00% |
| FLAVA | 99.74% | 99.24% | 97.76% | 98.22% | 100.00% |
| ALIGN | 99.74% | 100.00% | 99.44% | 99.75% | 100.00% |
| BLIP | 100.00% | 98.48% | 93.56% | 99.75% | 100.00% |
| average | $99.74\% \pm 0.26\%$ | $99.53\% \pm 0.50\%$ | $97.55\% \pm 2.09\%$ | $99.33\% \pm 0.54\%$ | $99.97\% \pm 0.08\%$ |

| Model | Accuracy | | | | |
| --- | --- | --- | --- | --- | --- |
| | parachute | golf ball | gas pump | garbage truck | chainsaw |
| CLIP:RN50 | 99.74% | 97.74% | 91.65% | 98.71% | 96.63% |
| CLIP:ViT-B/32 | 98.97% | 99.25% | 97.61% | 99.49% | 99.22% |
| CLIP:ViT-L/14 | 99.74% | 99.75% | 100.00% | 99.74% | 99.74% |
| OpenClip:ViT-B-32 | 99.49% | 99.25% | 97.61% | 99.23% | 97.67% |
| OpenClip:ViT-L-14 | 100.00% | 99.50% | 99.28% | 99.49% | 98.96% |
| FLAVA | 98.72% | 99.00% | 97.61% | 99.23% | 96.37% |
| ALIGN | 99.49% | 99.75% | 99.28% | 99.49% | 98.96% |
| BLIP | 99.49% | 99.50% | 98.09% | 99.23% | 98.19% |
| average | $99.46\% \pm 0.39\%$ | $99.22\% \pm 0.61\%$ | $97.64\% \pm 2.43\%$ | $99.33\% \pm 0.29\%$ | $98.22\% \pm 1.15\%$ |

## E.4 Imagenette Dataset

Here, we present additional information and results on the Imagenette dataset presented in the main body of this manuscript.[5] This dataset contains 13,394 images (9,469 for training and 3,925 for testing) from ten easily separable classes in ImageNet [22]. Fig. E.10 includes example images from the classes in the dataset, and Table E.6 reports the classification accuracy across all vision-language models ($98.99\% \pm 0.01\%$ average accuracy).

Recall that the ImageNet dataset does not provide ground-truth semantic annotations, hence we use SpLiCe [6] to find which concepts to test. In particular, we use the 10,000 most frequent words in the vocabulary from the MSCOCO dataset [40], and we set the $\ell_1$ regularization term in SpLiCe to 0.20. Following previous work [85], we filter the selected concepts such that they are different from the classes in the dataset. We use WordNet [23] to lemmatize both concepts and class names (e.g., "churches" becomes "church"), and we check that concepts are not contained in class names and vice versa. For example, the concept "churches" would be skipped because "church" is already the name of a class, and "gasoline" would be skipped because it contains part of the class "gas pump".

---

[5]The Imagenette dataset is available at https://github.com/fastai/imagenette.

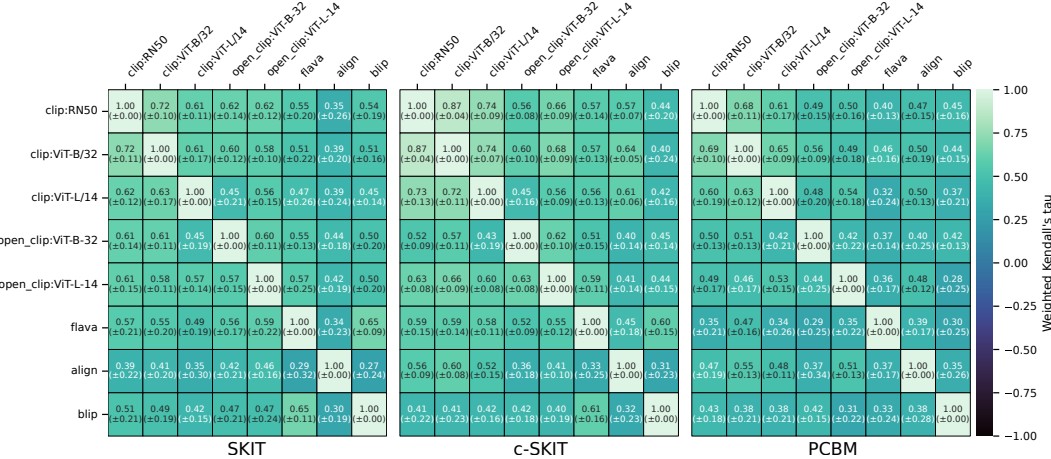

(a) Global importance with SKIT.

(b) Global conditional importance with C-SKIT.

Figure E.11: Global importance results with CLIP:ViT-L/14 on the Imagenette dataset.

Figure E.12: Rank agreement comparison between SKIT, C-SKIT, and PCBM on Imagenette. Result are reported as mean and standard deviation over the 10 classes considered in the experiment.

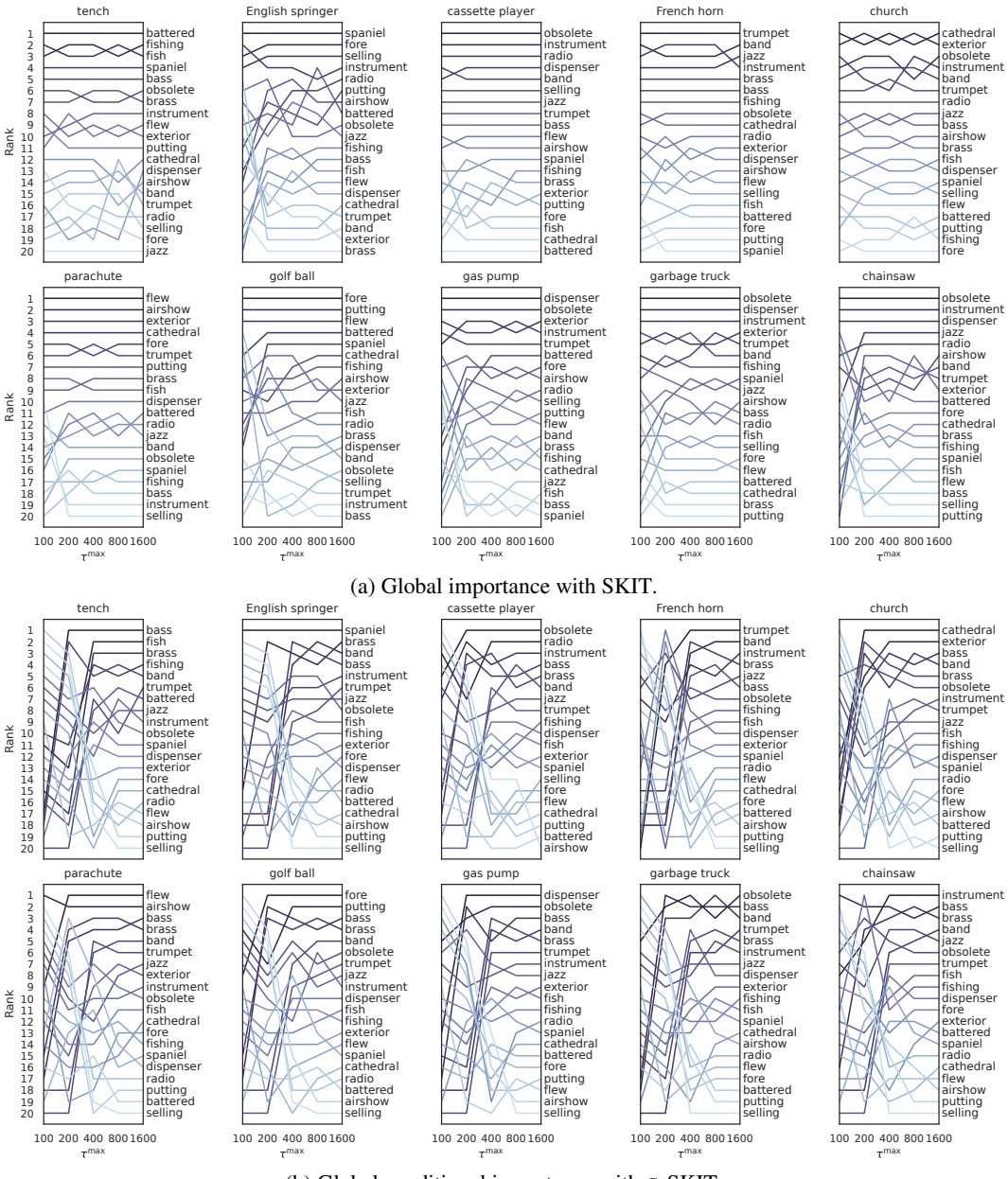

(a) Global importance with SKIT.

(b) Global conditional importance with C-SKIT.

Figure E.13: Importance results with CLIP:ViT-L/14 on Imagenette as a function of $\tau^{\max}$.

We run SKIT and C-SKIT with RBF kernels with bandwidths equal to the 90th percentile of the pairwise distances of previous observations and $\tau^{\max} = 400, 800$, respectively. We encode the entire dataset using SpLiCe and keep the top-20 concepts to test. Fig. E.11 shows global and global conditional importance results with CLIP:ViT-L/14 for all classes in the dataset. Furthermore, Fig. E.12 shows all pairwise rank agreement comparisons for SKIT, C-SKIT, and PCBM across all 8 VL models used in the experiment. Lastly, Fig. E.13 qualitatively shows ranks of importance as a function of $\tau^{\max}$ with CLIP:ViT-L/14.

We use X-SKIT on 2 random images from all classes in the dataset (20 images total). Recall that we use SpLiCe to encode each image and keep the top-10 concepts, and finally add the bottom-4 concepts according to PCBM, for a total of 14 concepts per image. We set the bandwidth of the RBF kernel used in the test to the median of the distance of the observations, and $\tau^{\max} = 200$. As in the CUB dataset experiment, we classify concepts as important by thresholding their rejection rates at

Table E.7: X-SKIT results on the Imagenette dataset as a function of conditioning set size $s$.

| $s$ | Rank agreement | Importance agreement |
|---|---|---|
| 1 | $\mathbf{0.59 \pm 0.21}$ | $\mathbf{0.71 \pm 0.14}$ |
| 2 | $0.56 \pm 0.21$ | $0.67 \pm 0.13$ |
| 4 | $0.53 \pm 0.23$ | $0.68 \pm 0.14$ |

(a) Rank agreement.

(b) Importance agreement.

Figure E.14: X-SKIT agreement results on the Imagenette dataset across 8 different vision-language models as a function of conditioning set size, $s$. Results are reported as means and standard deviations over the random 20 images used in the experiment.

level $\alpha$. Table E.7 summarizes rank and importance agreement as a function of conditioning set size $s$ (i.e., the number of concepts in $S$), and Fig. E.14 includes all pairwise agreement values. We can see that ranks are generally well-aligned across models, and that agreement slightly decreases as the number of conditioning concepts increases. Finally, Figs. E.15 and E.16 include results with X-SKIT for 2 images from three classes in the dataset across all models used in the experiment, and Fig. E.17 summarizes ranks across models on the same images.

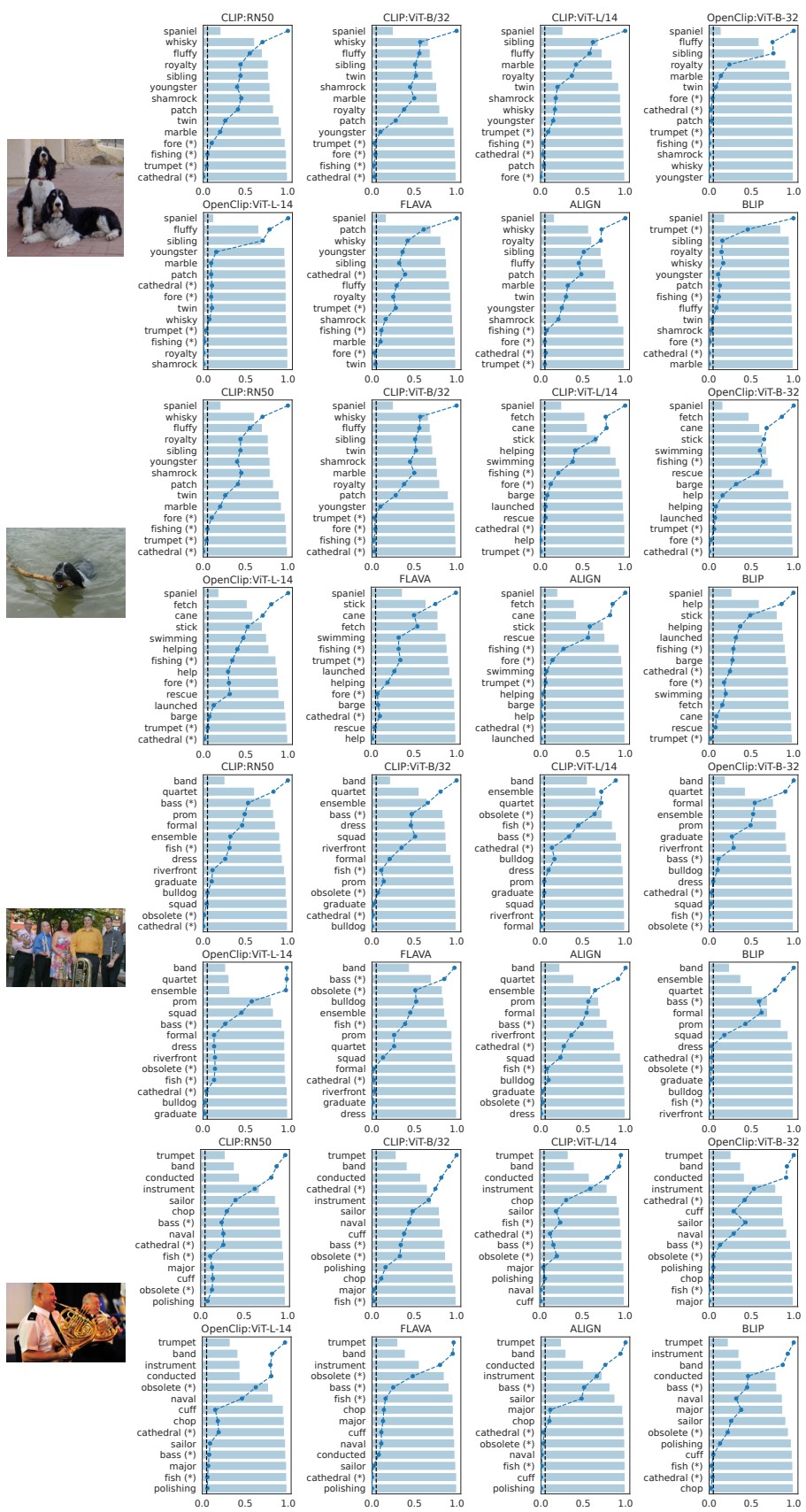

Figure E.15: Local importance results with SKIT and CLIP:ViT-L/14 for 2 images from three classes in the Imagenette dataset (part I of II.

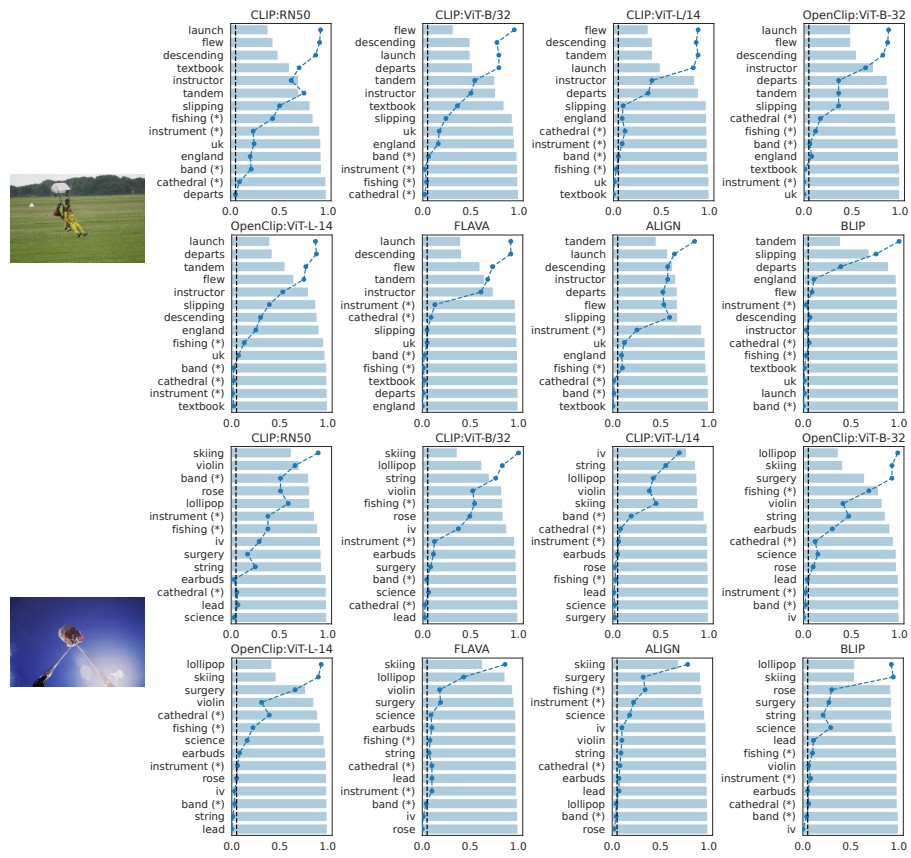

Figure E.16: Local importance results with SKIT and CLIP:ViT-L/14 for 2 images from three classes in the Imagenette dataset (part II of II).

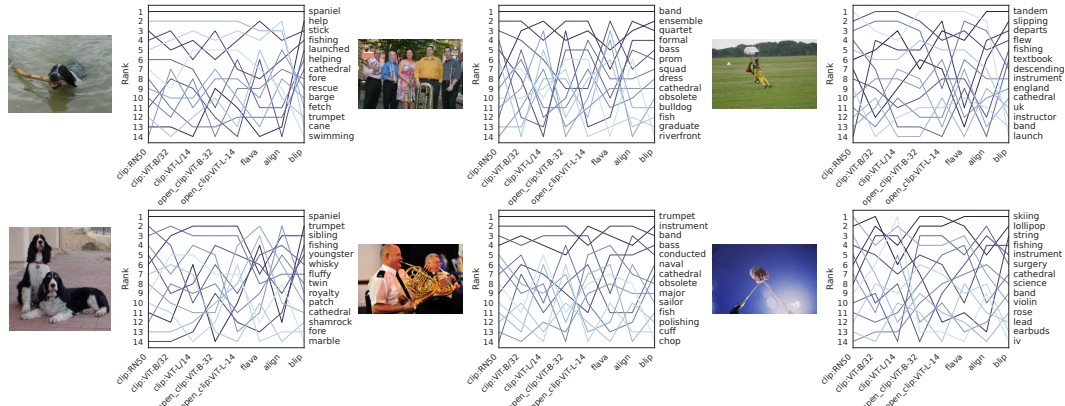

Figure E.17: Summary of X-SKIT ranks of importance across all models for 2 examples images from 3 classes in the Imagenette dataset.

