# OpenReview forum: "Testing Semantic Importance via Betting"
_NeurIPS.cc/2024/Conference — NeurIPS 2024 poster_

### Official Review · Reviewer_FuBo · 2024-07-09

**Soundness:** 2
**Presentation:** 3
**Contribution:** 4
**Rating:** 6
**Confidence:** 3

**Summary:**

The paper presents a method to test feature importance when a model makes its decision. The proposed method is based on hypothesis testing. The features concerned in this work is human-interpretable ones. For example, when CLIP makes a "cat" prediction to the image, the features tested are: "whiskers", "pointy ears", etc. The paper claims several novel contributions: (1) the method does not depend on the existence of dense features dictionary -- and by principle can take features as input from the users (2) the method keep the original predictor, unlike existing methods that train a separate predictor.

**Strengths:**

1. Conceptually very important and interesting, especially on the notion of sample-speficic vs. global vs global conditional
2. *if* really works on input features from user, is a very important contribution

**Weaknesses:**

Although the proposed method's efficacy and claims are very interesting, i find several major concerns:
1. the method part is very hard to understand -- especially on the equation part. I fins it hard to understand what does the equations mean, and how that connects to by rejecting the hypothesis, means that the prediction depends on the feature? the writing on method section can use a lot more explanation.
2. The evaluation is not convincing. The result presented claims that the feature importance outputted by the algorithm agrees with intuition -- this is very subjective. The evaluation can be significantly strengthen by human evaluation.

**Questions:**

Adding an actual "feature from user input" evaluation can significantly strengthen the claim. + see weakness

**Limitations:**

yes

---

> ### Author Rebuttal · Authors · 2024-08-06
>
> ## Thank you for your comments!
>
> We thank the reviewer for their encouraging comments. Here, we address each weakness point individually, and we are looking forward to discussing more.
>
> > *if* really works on input features from user, is a very important contribution.
>
> We do want to stress that our method *does* work with any set of concepts, and in particular it is the only method that can provide local semantic explanations for the predictions of a black-box model. So, we do appreciate the reviewer's comment that this is a very important contribution!
>
> ---
>
> ### **Clarity of presentation**
>
> Could the reviewer expand on their points of confusion? We would be more than happy to clarify methods and equations in the revised version of the manuscript.
>
> ---
>
> ### **Alignment with human intuition**
>
> With agree with the reviewer that a user study needs to be performed to claim alignment with human intuition. As we state in the general response at the top of our rebuttal, we will smooth such claims in the revised version of the manuscript. We note that our contributions *can* readily work with any set of concepts, including user-defined ones, and that no alternatives currently exist.
>
> **We have included several additional experiments to strengthen the experimental section of our submission and validate the ranks of importance obtained with our methods.**
>
> In particular, we have included the AwA2 and CUB datasets, which have ground-truth annotations of which concepts are present in images coming from certain classes or specific images, respectively. We refer the reviewer to our general response at the top of our rebuttal for a detailed description of the additional experiments and their results.
>
> To summarize our findings:
>
> - c-SKIT has better semantic importance detection performance compared to PCBM on AwA2.
> - x-SKIT has good alignment with ground-truth annotations across all models ($\approx 0.85$ average $f_1$ score) on CUB.

---

> > ### Comment · Reviewer_FuBo · 2024-08-07
> > **thank you for your response!**
> >
> > yes, i agree w/ the authors that the new results solidifies the claims. i have raised my score

---

> > > ### Author Response · Authors · 2024-08-09
> > > **Thank you for your response!**
> > >
> > > We sincerely thank the reviewer for engaging in discussion and their consideration of our rebuttal.
> > >
> > > We are glad to hear our additional experimental results solidified the claims in our manuscript.

---

### Official Review · Reviewer_nqVJ · 2024-07-11

**Soundness:** 3
**Presentation:** 3
**Contribution:** 3
**Rating:** 6
**Confidence:** 3

**Summary:**

Recently, there has been a lot of interest in understanding the inner workings of deep neural networks. Most existing works learn semantic concepts that are inherently understandable to the user. Often, each semantic concept comes with an associated score, and in many cases, it is hard to interpret the scores. Even though some works address this issue, they only work for input features and don't easily apply to semantic concepts. The work aims to address this shortcoming and formalizes the notion of statistical importance for both local and global semantic concepts.

**Strengths:**

- The work formalizes the notions of global, local, and global conditional importance.
- Most existing methods assume the presence of a large bank of concepts; the proposed method allows the stakeholders to specify the concepts they want to evaluate directly. This flexibility will enable explanations with diverse semantics for the model prediction on an example, as opposed to a single explanation.
- In practice, most of the methods rely on the weights of a linear model over the concepts to convey the importance of the semantic concepts. In contrast, since the proposed method utilizes statistical significance, it can guarantee false positive rates.
- Unlike existing methods, the proposed technique doesn't rely on training a surrogate linear model and can study the semantic importance of any given model.

**Weaknesses:**

The paper is generally sound regarding the contribution, but the authors should consider the questions below regarding the experiment.

- First, I encourage the authors to evaluate their technique on diverse tasks and datasets. For instance, they can evaluate their method on datasets like AwA2 and CUB, which already have concept annotations. In addition, the authors could assess their methods on domains like NLP.

- Why only consider a single backbone? I encourage the authors to include results from diverse backbones, strengthening the paper.

- One of the paper's main contributions is the ease with which an end-user can understand the semantic concepts and their importance, but that aspect still needs to be evaluated. The authors should design & conduct a user study to assess it.

- In addition, the authors can validate the concept's usefulness by intervening on the concepts and measuring the changes in model predictions. Another way to validate the concepts would be to measure their predictive ability; ideally, just using the important concepts to predict shouldn't hinder the model's performance.

**Questions:**

Refer to Weaknesses.

**Limitations:**

Yes, the authors discuss the limitations of the proposed technique in detail. They have sufficiently addressed it, and the contributions outweigh the limitations.

---

> ### Author Rebuttal · Authors · 2024-08-06
>
> ## Thank you for your comments and questions!
>
> We now address each point raised by the reviewer individually, and we are looking forward to clarifying any outstanding questions.
>
> ---
>
> ### **Diverse datasets and models**
>
> We thank the reviewer for their suggestions, which have significantly strengthen our experimental results and uncovered interesting findings.
>
> **As presented in the general response, we have now included additional experiments on both AwA2 and CUB-200-2011, comparing across 8 different models: CLIP:RN50, CLIP:ViT-B/32, CLIP:ViT-L/14, OpenClip:ViT-B-32, OpenClip:ViT-L-14, FLAVA, ALIGN, and BLIP.**
>
> In particular:
>
> - Since AwA2 includes class-level annotations, we have included global conditional importance results with c-SKIT in comparison with PCBM.
> - Since CUB includes image-level annotations, we have included local conditional importance results with x-SKIT.
>
> We describe all findings and experiments in the general response at the top of our rebuttal. To summarize, we find our c-SKIT outperforms PCBM both in terms of semantic importance detection and transferability across all models on both Imagenette and AwA2. Furthermore, x-SKIT importance ranks align well with ground-truth annotations ($\approx 0.85$ average $f_1$ score) on CUB.
>
> We consider NLP and language generation applications of our framework as important future directions. For example, it is still not completely clear how ideas of concept bottleneck models apply to encoders such as BERT. How do text encoders represent semantic information with standard pretraining techniques such as masked language modeling? Only very recent work [1] has started addressing these fundamental questions. Going beyond text encoders, how should one phrase questions of semantic importance for contrastive encoder-decoder architectures such as CoCa [2] or autoregressive models such as GPT? The structured nature of these models raises very important questions that currently remain unanswered, thank you for raising this point.
>
> We will include these discussion points in the revised version of the manuscript.
>
> ---
>
> ### **User study to evaluate alignment with human intuition**
>
> As stated in the general response, we agree with the reviewer that a user study would be necessary to claim alignment with human intuition, and we will rephrase those claims in the revised version of the manuscript. However, we remark that our tests *can* be readily applied to any set of concepts, including user-defined ones.
>
> We believe the design of a robust user study to deserve its own investigation, and, in this submission, we focus on introducing the methods that will enable such a study. We stress that, currently, no alternative method *can* work with a few user-defined concepts. For this reason, we envision our method to enable the design of such studies.
>
> ---
>
> ### **Validation of concept usefulness**
>
> We thank the reviewer for their suggestions.
>
> We validate important concepts in our additional experiments on AwA2 and CUB using the ground-truth annotations. We refer the reviewer to our general response and the attached pdf for numerical results and comparison with PCBM.
>
> Furthermore, we would like to mention that our c-SKIT and x-SKIT tests precisely work by resampling concepts and measuring their effect on distributions of the output of the predictor. However, we remark that we use *observational* conditional distributions (i.e., $Z_j \mid Z_{-j}$) and not *interventional* distributions (i.e., $Z_j \mid \text{do}(Z_{-j} = \cdot)$). Characterizing the connections between our framework and causal inference is an interesting future line of research, thank you for raising this point.
>
> Lastly, we kindly push back on the suggestion of validating concepts by training predictors, as our focus is to study which concepts are important for a fixed black-box model, and, in the general case, these may be different from the ones with highest predictive ability. This is a fundamental distinction between our approach and alternatives like PCBM. Intuitively, consider ImageNet classification. It might be the case that we want to test one concept only. But of course, even if that individual concept were truly used by the model, it will not be enough to train a good classifier from scratch. We can already see this behavior in the AwA2 experiments: restricting to 10 concepts, that we know from ground-truth annotations should be important, significantly reduces predictive power. We refer the reviewer to the "average" line in Table 1 of the attached pdf, where PCBM shows a drop in performance of around 4%.
>
> ---
>
> [1] Tan et al. "Interpreting pretrained language models via concept bottlenecks." (2024)
>
> [2] Yu et al. "Coca: Contrastive captioners are image-text foundation models." (2022)

---

> > ### Comment · Reviewer_nqVJ · 2024-08-12
> >
> > Thanks for the detailed response, and addressing my concerns, I will update my scores.

---

> > > ### Author Response · Authors · 2024-08-12
> > > **Thank you for your response!**
> > >
> > > We are glad to hear our response addressed the reviewer's concerns!
> > >
> > > We sincerely thank the reviewer for their comments and their consideration of our rebuttal, we will include all additional results in the revised version of the manuscript.

---

### Official Review · Reviewer_mYwv · 2024-07-13

**Soundness:** 3
**Presentation:** 3
**Contribution:** 3
**Rating:** 6
**Confidence:** 2

**Summary:**

The paper defines statistical importance of semantic concepts for black-box models such as CLIP via conditional independence.
This is motivated by the fact that users would be interested to know how they should interpret two concepts with different importance scores, and if the difference in such two concepts has any statistical significance.
The paper discusses that earlier work considers input features only, and not directly applicable to semantic concepts, which are represented in the internal layers of models.
The paper utilizes recent advances in sequential kernelized independence testing to develop statistical tests that can produce a rank of importance for each semantic concept.
The proposed method is experimentally verified using CLIP and a subset of ImageNet data.

**Strengths:**

- How to measure statistical significance between different semantic concepts' importance is well-motivated (e.g. "how should users interpret two concepts with different importance scores?", "Does their difference in importance carry any statistical meaning?" are valid questions that users would be interested to explore)
- Proposed two novel procedures to test for semantic importance: c-SKIT for global conditional importance, and x-SKIT for local conditional importance
- Validated their method on zero-shot ImageNet classification with CLIP.

**Weaknesses:**

While the proposed solution for discovering feature/concept importance of black-box models is written as a general framework, it's not clear how applicable this method is to problems in practice, because the real-data experiment is only performed on CLIP. For example, users in practice would probably be interested in seeing how much the rankings produced by the proposed method agree across CLIP and other vision-language models.

**Questions:**

CLIP has an issue with compositional understanding (e.g. Hsieh et al. "SugarCrepe", Sec 5.3), since the embedding space that is learned through CLIP's contrastive loss is incentivised to only match up to the set of concepts between images and texts, rather than learning the relational / compositional structure between concepts in the image / text. I'm wondering if the ranking order discovered by this method also suffers from this issue of CLIP's representations.

**Limitations:**

- Experiments on real datasets are limited, and it would be interested to know how much transferable the results of CLIP to other vision-language models.

---

> ### Author Rebuttal · Authors · 2024-08-06
>
> ## Thank you for your comments and questions!
>
> Here, we address each point individually, and we are looking forward to discussing with the reviewer.
>
> ---
>
> ### **Limited experiments and transferability across different vision-language models**
>
> This is a great question, and we thank the reviewer for raising this point. These additional experiments have strengthen the experimental results of our submission, and uncovered interesting aspects of transferability.
>
> **In particular, we have extended our experimental results to include 2 additional datasets (AwA2 and CUB) and 8 different models (both CLIP- and non-CLIP-based): CLIP:RN50, CLIP:ViT-B/32, CLIP:ViT-L/14, OpenClip:ViT-B-32, OpenClip:ViT-L-14, FLAVA, ALIGN, and BLIP.**
>
> We evaluate agreement between pairs of models in terms of ranking of concepts, and whether they are classified as important or not. To compare ranks, we use a weighted version of Kendall's tau (see [1]) which assigns higher penalties to swaps at higher positions. That is, for example, a 1 -> 5 swap is worse than a 4 -> 5 swap. This reflects that higher positions should matter more and be more stable. To compare importance, we threshold rejection rates at level $\alpha$ and compute the accuracy between the binarized vectors.
>
> We briefly summarize here the findings of our experiments, which are described in the general response:
>
> - Ranks obtained with c-SKIT are more transferable than PCBM on both Imagenette and AwA2.
> - Both CLIP- and non-CLIP-based models are generally aligned in terms of ranks and importance, especially on local explanations.
>
> ---
>
> ### **Compositional understanding of CLIP's embedding space**
>
> This is a very interesting point, which deserves further investigation outside of the current submission.
>
> For example, following the SugarCrepe example of a photo with *"a girl in white facing a man in black"*, one could test whether the prediction of *"girl"* does depend on the concepts *"white"*, *"facing"*; while the prediction of *"man"* depends on *"black"*. We stress that this study would not be possible without the tools presented in this submission.
>
> We would also note that our results on global (marginal) importance find that concepts are almost always important (i.e., their rejection rates are above $\alpha$). As suggested by the reviewer, this finding may support the claim that CLIP's semantic space is entangled and overlapping. These aspects have also been considered by previous works (for example, MERU [2]), which focus on whether using an embedding spaces different from the unit sphere (e.g., the hyperbole) induce better hierarchical representations. In a similar fashion to above, one could also devise an experiment to retrieve and test semantic image traversal as in [2, Figure 5].
>
> These example studies highlight that the framework presented in this submission has a broad reach beyond explainability and it will support other research efforts, thank you for raising this point.
>
> ---
>
> **References**
>
> [1] Vigna. "A weighted correlation index for rankings with ties." (2015)
>
> [2] Desai et al. "Hyperbolic image-text representations." (2023)

---

> > ### Comment · Reviewer_mYwv · 2024-08-11
> > **Response**
> >
> > Thank you for your detailed response. Additional experiments using different V&L models addressed my concern.
> > It is nice to see that ranks and importance are transferable across both CLIP and non-CLIP based models.
> > I increased my score to reflect these changes.

---

> > > ### Author Response · Authors · 2024-08-12
> > > **Thank you for your response!**
> > >
> > > We sincerely thank the reviewer for their consideration of our rebuttal.
> > >
> > > We agree that the findings on transferability would be valuable to readers, and we will include them in the revised version of the manuscript.

---

### Official Review · Reviewer_go3R · 2024-07-16

**Soundness:** 3
**Presentation:** 3
**Contribution:** 3
**Rating:** 6
**Confidence:** 3

**Summary:**

The paper discusses the need for precise statistical guarantees in feature importance, especially for semantic concepts, to ensure transparency and avoid unintended consequences.

It introduces a framework using conditional independence for testing semantic importance and demonstrates its effectiveness on synthetic datasets and image classification tasks using models like CLIP.

It uses principles of testing by betting (or sequential testing), which are based on e-values, and MMD as a test statistic

The paper finds importance of concepts via conditional independence

The authors introduce two novel procedures: conditional randomization SKIT (C-SKIT) for global conditional importance and explanation randomization SKIT (X-SKIT) for local conditional importance.

**Strengths:**

Offers rigorous definitions and tests for global and local semantic importance

It emphasizes the importance of interpretable features in black-box models

The paper is well-written and smoothly explains its argumentation.

**Weaknesses:**

The paper does little comparison to other state-of-the-art techniques for feature importance.

**Questions:**

The practical implementation assumes a small set of concepts. Do you find it to be enough?

 What are the computational limits of the proposed method?  How do they scale with increasing data size and complexity?

 For what data modalities is the given choice of kernel functional?

**Limitations:**

The practical implementation assumes a small set of concepts.

 For certain tests, accurate generative models for the conditional distributions are required, which can be difficult to train.

---

> ### Author Rebuttal · Authors · 2024-08-06
>
> ## Thank you for your comments and questions!
>
> We address each point individually, and we are looking forward to discussing with the reviewer to answer any outstanding questions.
>
> ---
>
> ### **Comparison with SOTA**
>
> We thank the reviewer for this comment, which has significantly strengthened our experimental results. We have described several additional experiments and comparisons with PCBM (which is the state-of-the-art for semantic explanations) in the general rebuttal. To summarize, we find that:
>
> - c-SKIT provides ranks that are more transferable across different vision-language models on both Imagenette and AwA2.
> - c-SKIT provides better semantic importance detection in terms of $f_1$ score on AwA2.
>
> For local semantic explanations, we have included additional experiments on CUB-200-2011, which indicate x-SKIT is well-aligned with ground-truth ($\approx 0.85$ average $f_1$ score). We note that, currently, there are no alternative methods that can produce local semantic explanations, hence why we could not compare.
>
> For the sake of completeness, for global conditional importance, we tried comparing with LaBo (Yang et al., 2023), which, intuitively, adds a softmax activation to the weights of a PCBM classifier. This approach, however, fails to learn good predictors in our few-concepts setting ($\approx 20\\%$ classification accuracy on AwA2 dataset compared to $\approx 95\\%$ for PCBM).
>
> ---
>
> ### **Small number of concepts**
>
> We are not sure we fully understand the question. Could the reviewer expand on what they mean by *``enough''*?
>
> Our proposed framework does not modify the original black-box predictor, whose accuracy does not depend on the number of concepts. So, defining a small set of concepts does not affect the performance of the predictor.
>
> On the other hand, it is true that the number of concepts may thwart the ability to build effective samplers to instantiate our c-SKIT and x-SKIT tests. For example, in the case of limited data, it may not be feasible to train a conditional sampler on a large set of concepts. In our experiments, we focus on $\approx 20$ concepts because previous work [1] has shown that humans prefer succinct explanations. We found that this assumption allows us to use non-parametric samplers which are fast, cheap, and do not require prior training.
>
> We will clarify this in the revised version of the manuscript, thank you.
>
> ---
>
> ### **What are the computational limits? How do the tests scale with increasing data size and complexity?**
>
> We thank the reviewer for these questions.
>
> First, the main computational limit is the need of conditional samplers. In certain domains, these models may be expensive both to train and run, such is the case for diffusion or language models. In our experiments, we strived to use methods that are effective but do not require prohibitive computational resources.
>
> Second, the computational complexity depends on the specific test and the sampler. In particular:
>
> - SKIT: Following [2, Appendix F.2] the test runs in $O(\tau^2)$, where $\tau$ is the (random) stopping time of the test. This is because at each step $t$, computing the MMD requires summing over the previous $t-1$ terms.
> - c-SKIT and x-SKIT: both tests incur in a extra factor of $T_n$, which represents the cost of the sampler on $n$ data points.
>
> Finally, we note that the runtime does not depend on the number of concepts because different concepts can be tested simultaneously. We will include this in the revised version of the paper.
>
> ---
>
> ### **For what data modalities is the given choice of kernel functional?**
>
> This is a great point!
>
> Our framework assumes the black-box predictor can be divided into an encoder and a classifier, and these need to be appropriate for the data modality at hand. Once inputs are mapped to an embedding space, one needs to decide on which kernel to use to test for semantic importance.
>
> In general, when using kernel methods to test for a null hypothesis, we would like the kernel to be *characteristic* for the alternative (see [2]). That is, we want the kernel to be expressive enough to distinguish two distributions coming from the alternative. Over compact domains, this can be achieved by using *universal* (see [3]) kernels---such as the RBF kernel.
>
> In practice, this means that:
>
> - for x-SKIT, the RBF kernel is appropriate whenever the classifier is a real-valued function (e.g., linear classifier with sigmoid activation). For discrete predictors (e.g., decision trees), other kernels may be necessary.
> - for c-SKIT and SKIT, the RBF kernel is appropriate whenever both the predictor and the concept bottleneck layer are real-valued functions (e.g., linear classifier with sigmoid activation and SVMs, respectively).
>
> We remark that our tests are defined for any choice of kernel, and they can be instantiated directly for the desired data modality.
>
> Finally, we refer the reviewer to Fig. E.3 in the Appendix, where we precisely compare c-SKIT with a linear and RBF kernel on a synthetic dataset. This experiment shows that a linear kernel may fail to detect important concepts because it does not satisfy the universality property.
>
> ---
>
> **References**
>
> [1] Ramaswamy et al.  "Overlooked factors in concept-based explanations: Dataset choice, concept learnability, and human capability." (2023)
>
> [2] Podkopaev et al. "Sequential Kernelized Independence Testing." (2023)
>
> [3] Sriperumbudur et al. "Universality, Characteristic Kernels and RKHS Embedding of Measures." (2011)

---

### Author Rebuttal · Authors · 2024-08-06

## Thank you for your comments!

We sincerely thank all reviewers for their valuable comments and suggestions, which have strengthen our experimental results and the presentation of our contributions.

**Following all reviewer comments, we have significantly extended our experiments on real world data: we now include additional results on both the AwA2 and CUB datasets, and we compare across 8 different vision-language models.**

In this general response, we address common questions raised by several reviewers, and briefly summarize additional results presented in the rebuttal pdf. We address comments from each reviewer in their individual response, and we are looking forward to discussing with the reviewers to answer any outstanding questions.

---

## Imagenette results on different models

**As suggested by RmYwv, RnqVJ, and Rgo3R, we have extended our analysis to include 8 different models: CLIP:RN50, CLIP:ViT-B/32, CLIP:ViT-L/14, OpenClip:ViT-B-32, OpenClip:ViT-L-14, FLAVA, ALIGN, and BLIP.**

We evaluate agreement between all pairs of models with the following metrics:

- *Comparison of ranks.* We use the method of [1] to compare ranks. A value of -1 means reverse order, and +1 means perfect alignment.

- *Comparison of importance.* We threshold rejection rates at level $\alpha$ to classify concepts as important or not. Importance agreement is the accuracy between the binarized vectors.

We summarize the results included in the pdf:

- Fig. 1, average global importance agreement equal to 0.57 (random baseline = 0.00).
- Fig. 2, comparison of c-SKIT and PCBM. **We find c-SKIT ranks to have higher agreement (0.64) compared to PCBM (0.52), which indicates c-SKIT is more transferable than PCBM.**
- Fig. 3, average local conditional importance agreement is 0.73 and rank agreement is 0.62. **These results indicate that different vision-language models share certain local semantic dependence structures.**

---

## Global conditional importance on AwA2

As suggested by RnqVJ, we include global conditional importance results on the AwA2 dataset. We use c-SKIT on the top-10 best classified classes for a fair comparison across models. For each class, we test 20 concepts: 10 present, and 10 absent according to the ground-truth annotations.

We compute rank agreement and report $f_1$ scores between the ground-truth annotations and the top-10 concepts according to c-SKIT rejection times and PCBM absolute weights. We remark that the coefficients of a linear classifier are a heuristic notion of global conditional independence, whereas our tests provide precise statistical guarantees.

We summarize the results included in the pdf:

- Fig. 4, comparison of ranks obtained with c-SKIT and PCBM. **Similarly to above, we find c-SKIT ranks to be more transferable than PCBM's (0.54 vs 0.44 agreement)**.
- Table 1, $f_1$ scores for c-SKIT and PCBM. **c-SKIT consistently outperforms PBCM across all models (0.55 vs 0.48 average $f_1$ score). We stress this improvement in semantic importance detection does not reduce classification accuracy (99.50% vs 95.10%)**.

---

## Local importance on CUB

As suggested by RnqVJ, we include local importance results on the CUB-200-2011 dataset. We use x-SKIT on 2 test images from the top-10 best classified classes for a fair comparison across models. For each image, we test 14 concepts: 7 present, and 7 absent according to the ground-truth annotations. We threshold rejection rates at level $\alpha$ to classify concepts as important or not.

- Fig. 5, rank and importance agreement. **We find an average importance agreement of 0.97 and an average rank agreement of 0.86.**
- Table 2, $f_1$ scores as a function of size of conditioning set, $s$. **We find an average $f_1$ score of $0.84, 0.86, 0.83$ for $s \in \\{1,2,4\\}$. OpenClip:ViT-L/14 has the highest $f_1$ scores across all values of $s$, with a maximum of 0.89 for $s=2$.**

These results suggest models are aligned well with the ground-truth annotations.

- Fig. 6, example image with local importance ranks across different models.

---

## User study to evaluate alignment with human intuition

We agree with RnqVJ and RFuBo that a user study would be necessary to claim alignment of explanations with human intuition. **We will smooth these claims in the revised version of the paper in order to highlight that the scope of this work is to introduce a statistically-rigorous method that *can* work with any set of user-defined concepts**.

Finally, we would like to remark that, currently, there are no alternatives to design such a study but our framework. In fact, we envision many studies to leverage the precise statistical guarantees provided by our methods.

---

## FDR control

For the sake of completeness, we have also addressed FDR control limitations, as we mentioned in the submitted manuscript. We have extended our results to report important concepts with FDR control at level $\alpha$. We will include this in the revised version of the manuscript.

---

**Details**

AwA2 classes: giant panda, tiger, giraffe, zebra, lion, squirrel, sheep, horse, elephant, dalmatian.

CUB classes: White Pelican, Brown Pelican, Mallard, Horned Puffin, Vermilion Flycatcher, Northern Flicker, Cardinal, Blue Jay, Cape Glossy, Starling, Frigatebird.

**References**

[1] Vigna. "A weighted correlation index for rankings with ties." (2015)

---

### Decision · Program_Chairs · 2024-09-25

**Decision:**

Accept (poster)

**Comment:**

All reviewers unanimously recommended acceptance. It is a neat, well-written work that adds value to the literature on interpretability. The new experiments provided in the rebuttal have additionally improved the paper a lot, so please make sure to include them in the camera-ready version.